# Systematic prioritization of functional variants and effector genes underlying colorectal cancer risk

Philip J. Law [1], James Studd [1], James Smith[1], Jayaram Vijayakrishnan [1], Bradley T. Harris [2,3], Maria Mandelia [1], Charlie Mills [1], Malcolm G. Dunlop [2] & Richard S. Houlston [1] ✉

Genome-wide association studies of colorectal cancer (CRC) have identified 170 autosomal risk loci. However, for most of these, the functional variants and their target genes are unknown. Here, we perform statistical fine-mapping incorporating tissue-specific epigenetic annotations and massively parallel reporter assays to systematically prioritize functional variants for each CRC risk locus. We identify plausible causal variants for the 170 risk loci, with a single variant for 40. We link these variants to 208 target genes by analyzing colon-specific quantitative trait loci and implementing the activity-by-contact model, which integrates epigenomic features and Micro-C data, to predict enhancer–gene connections. By deciphering CRC risk loci, we identify direct links between risk variants and target genes, providing further insight into the molecular basis of CRC susceptibility and highlighting potential pharmaceutical targets for prevention and treatment.

CRC, which affects around 1.9 million people worldwide annually, has a strong heritable basis[1]. Our recent genome-wide association study[2] (GWAS) of 100,204 CRC cases and 154,587 controls has identified over 200 statistically significant independent risk loci. Deciphering the functional basis of these risk associations has the potential to provide biological insights into the etiology of CRC. However, deconvolution of GWAS risk loci has proven challenging owing to linkage disequilibrium between variants, and because most risk variants localize to noncoding regions of the genome, particularly within enhancer elements. Computational fine-mapping approaches can only predict putative causal variants based on linkage disequilibrium correlations[3]. To definitively identify variants with gene regulatory effects requires experimental validation.

Most noncoding GWAS risk variants are likely to function through *cis*-regulatory mechanisms that influence target gene expression. By investigating the transcriptional changes associated with different variants, it is possible to link specific alleles to changes in gene expression. Classical reporter assays can only assess the allelic transcriptional activity of individual variants. By contrast, massively parallel reporter assays (MPRAs) provide a scalable approach to characterize the regulatory effects of thousands of variants[4], and this strategy has recently been successfully exploited in studies to implicate variants associated with multiple disease states[5], including myeloma[6] and melanoma[7,8].

Although advances in fine-mapping and functional annotation of risk loci have improved the nomination of causal variants, identifying target genes for GWAS signals remains a central challenge. Traditionally, variants have been assigned to the closest gene. However, solely relying on physical proximity for prediction can be unreliable, as causal variants are often regulatory and can affect gene expression through long-range interactions[9,10]. Furthermore, it is now recognized that enhancers can have more than one target gene[11]. The analysis

[1]Division of Genetics and Epidemiology, The Institute of Cancer Research, Sutton, UK. [2]Colon Cancer Genetics Group, Edinburgh Cancer Research Centre, Institute of Genetics and Cancer, University of Edinburgh, Edinburgh, UK. [3]Present address: Wellcome Sanger Institute, Hinxton, UK. ✉e-mail: richard.houlston@icr.ac.uk

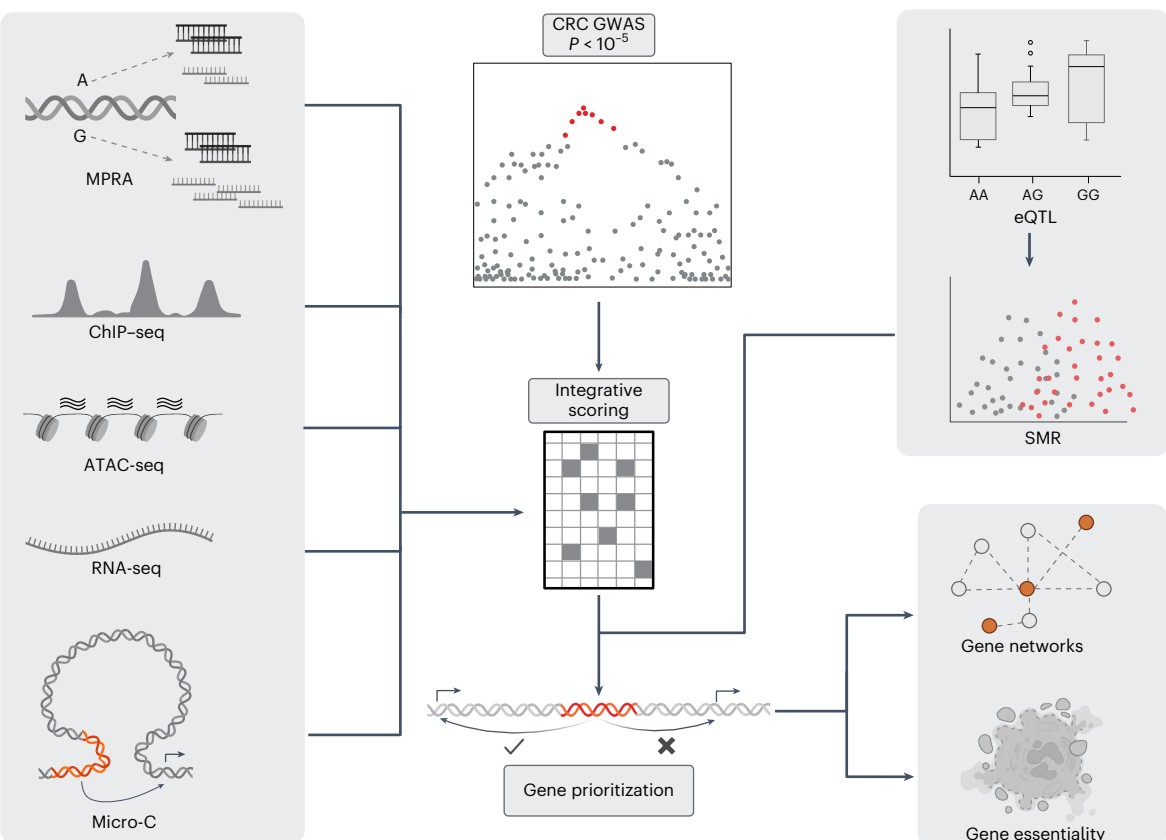

**Fig. 1 | Overview of the study.** Using data from GWASs for CRC, we identified 170 regions of interest. Data from MPRAs, epigenetic marks (ChIP–seq), chromatin accessibility (ATAC-seq), gene expression (RNA-seq) and long-range chromatin interactions (Micro-C) were combined to derive an integrative score to prioritize the functional variants at each CRC risk locus. These variants were linked to target genes by analyzing colon-specific eQTLs and using SMR. In the GWAS plot, the coloured dots indicate the variants that are above the P value threshold. In the SMR plot, they represent the two different datasets (GWAS and eQTL). The coloured portions of DNA represent the genomic regions of interest that were studied.

of expression quantitative trait locus (eQTL) data generated across multiple cell types has undoubtedly greatly aided target gene identification. However, because published eQTLs capture only 9–13% of the GWAS heritability of cancers[12], genomic data beyond gene transcription are required to comprehensively decipher the functional basis of associations[13]. Chromatin interactions and their proximity in genomic space are important for the regulation of gene expression. The integration of data from chromatin accessibility[14], epigenomics histone ChIP-seq (chromatin immunoprecipitation followed by sequencing)[15] and three-dimensional (3D) chromatin interactions[16–18] has been shown to improve the ability to identify causal variants and their likely target gene. The recognition of the limitations of reliance on a single analysis to identify causal variants and gene targets underlying GWAS signals has led to the adoption of data integration approaches[7,8]. For example, the INQUISIT pipeline, which scores gene expression, chromatin interactions and ChIP–seq annotations, has frequently been adopted by breast cancer researchers to identify candidate gene targets[19,20]. More recently, the computational approach implemented in the activity-by-contact (ABC) model has sought to systematically link regulatory elements to target genes through the combination of enhancer activity with 3D chromatin contact frequencies[21,22].

To provide insight into the functional basis of the CRC risk loci, we integrated data from multiple data modalities. First, we nominated causal variants at each of the risk loci through statistical fine-mapping incorporating tissue-specific epigenetic annotations, and by performing MPRAs in multiple colonic cell lines. Second, by generating and analyzing tissue-specific gene expression data and high-resolution chromatin interaction profiles, we linked nominated variants to target genes (Fig. 1 and Extended Data Fig. 1). Our analyses provide a detailed interpretation of CRC risk signals and their underlying basis.

## Results

### Cell specificity and chromatin landscape at risk loci
To identify the cellular contexts of the CRC loci, we analyzed single-cell RNA sequencing (scRNA-seq) profiles across 24 different tissues using the Tabula Sapiens dataset[23], as well as 11 intestinal regions in the Gut Cell Atlas[24]. We derived single-cell disease relevance scores (scDRSs), which link the scRNA-seq data with polygenic disease risk at single-cell resolution. This score assesses cell-type-specific expression for genes implicated by the GWAS association statistics. Genes whose expression was correlated with scDRSs were strongly enriched in large intestine and epithelial tissue ($P < 10^{-7}$). A specific analysis of intestinal cells showed a strong association of risk variants with BEST4$^+$ enterocytes and colonic epithelial cells ($P < 10^{-7}$, Supplementary Fig. 1). GWAS variants are generally thought to influence risk through *cis*-regulatory mechanisms affecting tissue-specific gene expression. We confirmed significant enrichment of enhancer- and promoter-associated histone marks, including histone H3 lysine 4 trimethylation (H3K4me3), H3 lysine 4 monomethylation (H3K4me1) and H3 lysine 27 acetylation (H3K27ac) in colonic and rectal mucosa cells using ChIP–seq data from the National Institutes of Health (NIH) Roadmap Epigenomics Project[25] ($P < 10^{-5}$, Supplementary Fig. 2).

### Statistical fine-mapping of risk loci
We fine-mapped each of the risk loci, including independent signals, incorporating functional annotation using PolyFun[26] and susieR[27] in

conjunction with ChIP–seq data on H3K4me1, H3K4me3, H3K27ac, H3K27me3, H3K36me3 and CCCTC-binding factor (CTCF) marks, as well as assay for transposase-accessible chromatin with sequencing (ATAC-seq) data generated from six CRC cell lines (C32, CL11, HT29, SW403, SW480 and SW948) (Supplementary Table 1). For each independent risk locus, we extracted variants within a 1-Mb window and calculated the causal probabilities nonparametrically using the established PolyFun protocol, which estimates the per-single nucleotide polymorphism (SNP) heritability weighted by their functional annotations. Credible sets of causal variants were identified by susieR using the probabilities calculated by PolyFun. Posterior inclusion probabilities (PIPs) were ranked, and variants were added to the set until the cumulative PIP reached a value of >0.95, with a minimum individual variant PIP of 0.001. We identified 1–14 credible sets per locus (median, 1), consisting of 1–226 variants (median, 1) (Supplementary Table 2).

### Functional significance of risk variants

We next assessed the regulatory activity of variants at each of the risk loci using a complementary experimental approach. At each GWAS locus (defined by a 500-kb window spanning the lead variant), we initially identified all variants with a $P$ value within three orders of magnitude of the $P$ value of the lead variant. As this may exclude potentially functional variants at loci where the lead variant has an especially strong association, we also included variants with $-\log_{10}(P_{variant}) > 0.7 \times (-\log_{10}(P_{lead\ variant}))$, stipulating an $r^2$ of >0.2 for the lead variant and a $P_{variant}$ of <$10^{-5}$. We performed MPRAs to simultaneously identify functional $cis$-regulatory variants, testing 8,880 variants (median of 39 variants per locus).

We evaluated the enhancer activity of reference and alternative alleles of the variants by cloning the surrounding 200 bp of genomic sequences. To test variant function in cellular constructs representing tumor and normal states, we transfected primary CRC cell lines (HT29 and SW403) and an immortalized primary colonic cell line (HCEC-1CT). Enhancer activity was quantified by sequencing barcodes in input DNA and mRNA (cDNA). Sequencing statistics and details of the quality control process are shown in Supplementary Table 3 and Supplementary Fig. 3. A total of 275 unique variants displayed significant allelic transcriptional activity (false discovery rate (FDR) < $10^{-3}$; $n$ = 133 in HT29, $n$ = 102 in SW403 and $n$ = 143 in HCEC-1CT; Supplementary Table 4). These 'MPRA-significant' variants were more likely to be fine-mapped as the causal variant (chi-square test, $P$ = 4.39 × $10^{-3}$) as well as fine-mapped to enhancer and promoter regions of the colonic epigenome ($P$ = 3.66 × $10^{-18}$).

We focused on the underlying biological mechanisms through which genetic variants at CRC risk loci shape the regulatory environment around putative target genes. First, because risk variants can mediate their effects through altered transcription factor binding, we assessed transcription factor binding in chromatin-accessible regions using the JASPAR 2022 transcription factor motif database[28] in concert with the ATAC-seq data. The most common transcription factors predicted to bind at the loci included ZNF460 (found at ten loci), CTCF ($n$ = 7), PRDM9 ($n$ = 7), SP1 ($n$ = 7) and KLF5 ($n$ = 3), and these transcription factor binding sites were enriched at the GWAS loci ($P$ < $10^{-4}$, Supplementary Fig. 4). Of note is KLF5, which was associated with the 13q22.1 risk locus, and PRDM9, a histone methyltransferase, which catalyzes H3K4 methylation. Second, we predicted enhancer–gene connections across risk loci from ultra-high-resolution Micro-C chromatin interaction profiles generated in CL11, HT29, SW403, SW403 and SW498 cell lines. The MPRA-significant variants preferentially localized to open chromatin ($P$ = 7.32 × $10^{-35}$) and mapped to regions that interacted with the transcription start site (TSS) of genes through a Micro-C chromatin interaction ($P$ = 7.28 × $10^{-4}$). In addition to confirming the interaction between rs6983267 at the 8q24.21 locus and the *MYC* TSS[18,29,30], chromatin looping interactions implicate several other genes with established roles in CRC biology, including *LAMC1*,

*TGFB1* and *KLF5*. Using Akita[31], a convolutional neural network based model for predicting 3D chromatin structure, 20% (1,798 out of 8,880) of the tested variants were predicted to affect 3D genome folding; 244 variants mapped to a CTCF motif, and approximately half of these ($n$ = 121) were predicted to severely affect the 3D chromatin structure.

### Nominating causal variants using an integrative scoring system

To prioritize plausible causal variants at each locus, we integrated the multiple levels of functional annotations and fine-mapping data for all 8,880 variants. We adopted a scoring approach similar to that of ref. 8, assigning a value between 0 and 2 for each variant and each annotation: 0 represented no hit, 1 represented a hit and 2 represented a strong hit (see Methods for the designation of each annotation-specific score). For each locus, the annotation scores were summed, and the variants ranked. The variants with scores in the top 20% were designated as Tier 1 variants, those with scores in the bottom 50% as Tier 3, and the remainder as Tier 2 (Fig. 2). We identified 2,406 Tier 1 variants, 42 of which were also the top hit in the GWAS meta-analysis. Forty-nine of the GWAS loci did not have any Tier 1 variants, and 16 of these also did not have any Tier 2 SNPs, with the 16 corresponding to regions with little to no functional data (Supplementary Fig. 5 and Supplementary Table 2).

### Linking nominated variants to target genes

To link variants at each locus to respective target susceptibility genes, we analyzed eQTL data from normal colon (SOCCS (Study of Colorectal Cancer in Scotland) colon or rectum epithelium, $n$ = 213; Genotype-Tissue Expression (GTEx) transverse colonic mucosa, $n$ = 367) and CRC tissues (The Cancer Genome Atlas Colon Adenocarcinoma (TCGA COAD), $n$ = 286; Rectum Adenocarcinoma (READ), $n$ = 94). Of the 275 MPRA-significant variants, 113 had a significant eQTL ($P_{eQTL}$ < 7.51 × $10^{-5}$; Bonferroni-corrected $P$ value for the 665 unique genes tested in the eQTL analysis), and 79 of these displayed a consistent direction of effect between MPRAs and eQTLs (that is, a direction of gene expression that is concordant with MPRA-allelic transcriptional levels) (Supplementary Table 2). By performing a summary-data-based Mendelian randomization (SMR) analysis[32], we identified 94 candidate target genes for 54 risk loci ($P_{SMR-adjusted}$ < 0.05; a median of one gene per locus) in the normal data, and 14 candidate target genes for 12 of the risk loci in the tumor data (Supplementary Table 5).

Following on, we evaluated the quantitative effect of enhancer–gene regulation by analyzing Micro-C data in conjunction with ATAC-seq, H3K27ac ChIP–seq and RNA-seq data using the ABC tool[22]. ABC interactions typically regulated two to three genes within 15–54 kb, and 62 of the risk loci fell within predicted enhancer regions that regulate genes.

Focusing on the Tier 1 variants at each locus (1–5 variants per locus; median, 1), 94 of the GWAS loci were linked to genes predicted by at least two sources of evidence (SMR normal, SMR tumor, ABC and Micro-C), and 10 had one source of evidence (Supplementary Table 6). Forty-two loci could not be associated with a gene, with the majority of these falling in expression-inactive regions (B-compartments). For 82 of the nominated loci, the closest gene was predicted to be the target gene, with 61 of these falling within introns. Approximately 70% of the nominated variants fell within the same topologically associating domain (TAD) as their target gene.

In addition to validating rs6983267, which mediates its effect through a long-range interaction with *MYC*[18,29,30], as the basis of the 8q24.21 association (Fig. 3a and Extended Data Fig. 2), our analysis provides evidence for the functional basis of the 170 risk loci and implicates 208 target genes. Although many of the risk loci have not previously been the subject of detailed scrutiny, several of the target genes have either well-documented roles in CRC or are strong a priori candidates for having a role in tumor biology. For example, we identify rs1248418 as the basis of the 10p12.1 association (top GWAS variant rs1773860;

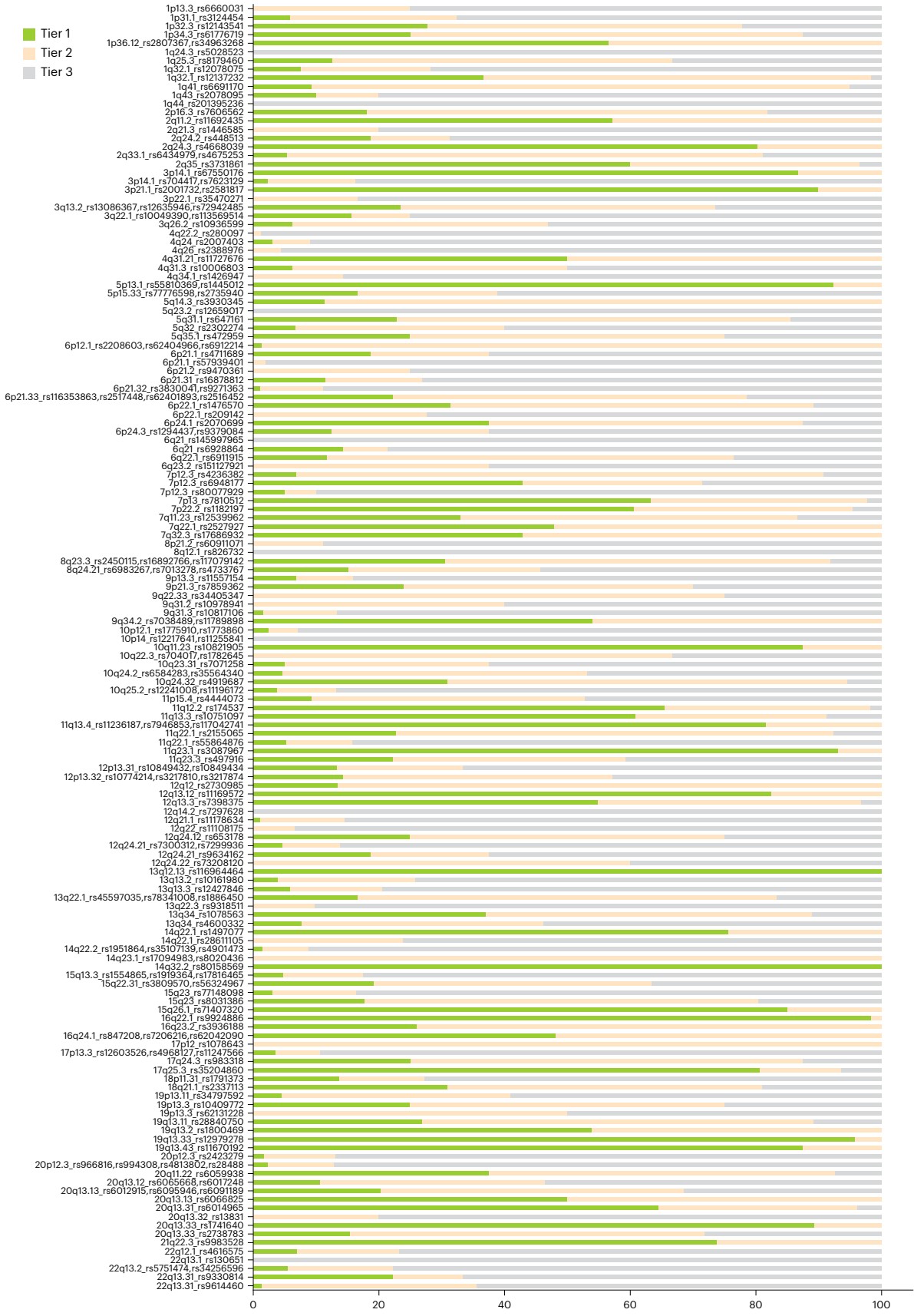

**Fig. 2 | Distribution of annotation scores for each GWAS locus.** Scores were calculated as the sum of the annotations for each variant. Loci are labeled with the cytoband and the top GWAS SNPs in each region. The variants with scores in the top 20% were designated as Tier 1 variants, those with scores in the bottom 50% as Tier 3 and the remainder as Tier 2.

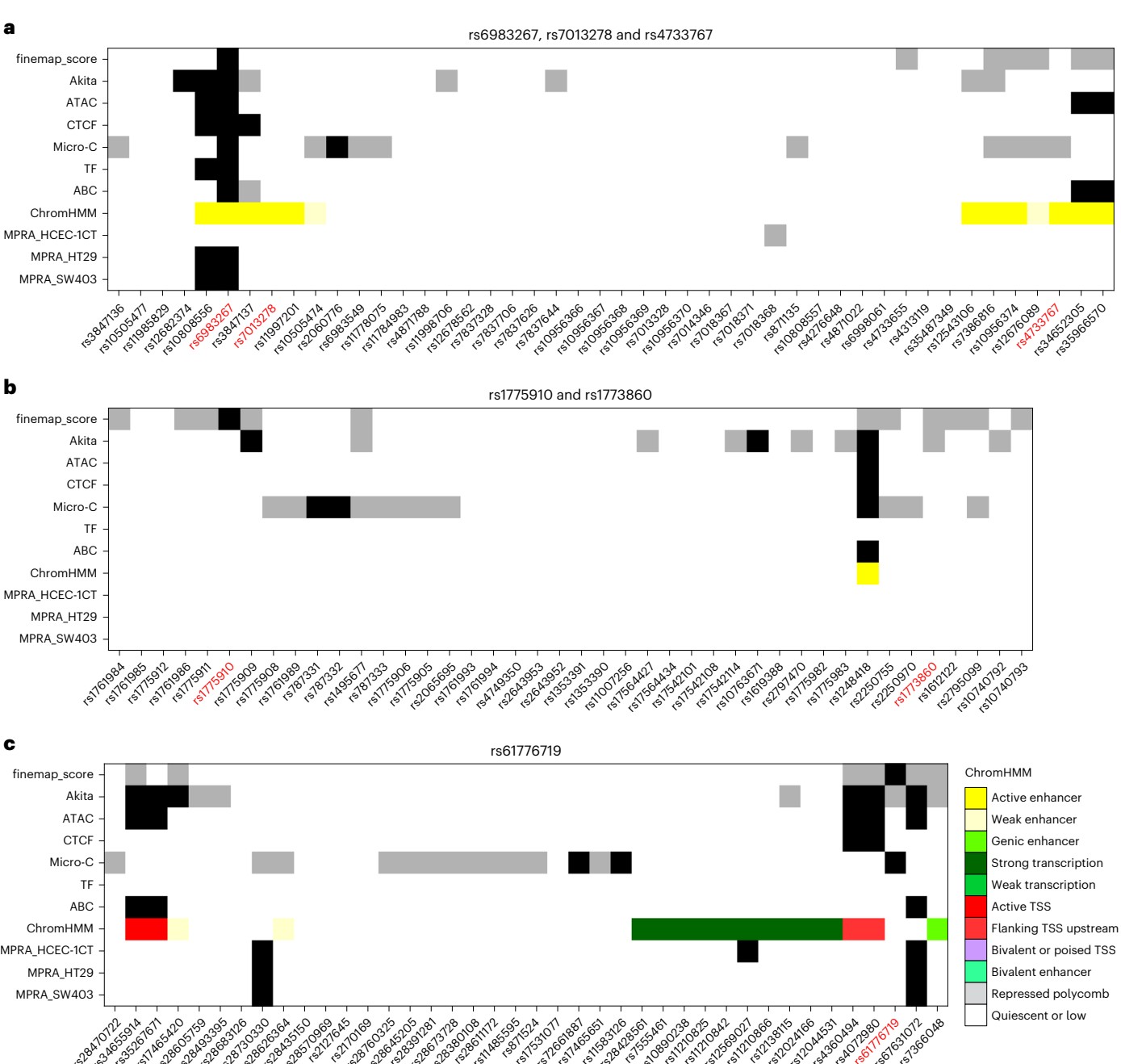

**Fig. 3 | Plot of the annotation sources for each of the variants analyzed in each GWAS locus. a**, At the 8q24.21 locus, the GWAS identified rs6983267, rs7013278 and rs4733767, highlighted in red, as risk loci. rs6983267 and rs7013278 are within 1.5 kb of each other, but rs6983267 is better annotated, with strong hits for MPRAs, transcription factor binding, open chromatin (ATAC-seq) and Micro-C. rs4733767 is over 150 kb away from rs6983267 and rs7013278 and has separate annotations, so it is probably a true independent locus. **b**, At the 10p12.1 locus, rs1773860 was the lead GWAS variant at this locus, but rs1248418 ($r^2 = 0.91$, $D' = 0.98$) was better annotated. This variant is located in open chromatin and is predicted to be in an enhancer region. In addition, this variant showed

a long-range interaction with the TSS of *BAMBI*. **c**, Functional annotation of rs61776719 at the 1p34.3 locus identified rs67631072 ($r^2 = 1.0$, $D' = 1.0$) as the top annotated variant, which shows enhancer activity in open chromatin regions and is predicted by the ABC model to affect gene expression. Detailed figures of the annotations of the regions are shown in Extended Data Figs. 2–4. In all figure panels, gray blocks correspond to an annotation, and black blocks correspond to a strong annotation. ATAC denotes the presence of an ATAC-seq peak, CTCF denotes the presence of a CTCF peak from the ChIP–seq analysis and Akita denotes evidence of disruption of 3D chromatin structure. TF denotes that a transcription factor was predicted to bind.

$r^2 = 0.91$, $D' = 0.98$). The enhancer region to which rs1248418 localizes shows a long-range interaction with the TSS of the gene encoding BAMBI, a negative regulator of transforming growth factor-β (TGFβ) signaling (Fig. 3b and Extended Data Fig. 3). Similarly, through functional annotation, we identify rs67631072 as the basis of the 1p34.3 locus (top GWAS variant rs61776719; $r^2 = 1.0$, $D' = 1.0$), with evidence from SMR and Micro-C implicating *FHL3* (Fig. 3c and Extended Data

Fig. 4). The C-risk allele ($P_{variant} = 1.59 \times 10^{-16}$) is associated with increased expression of *FHL3* ($P_{eQTL} = 7.69 \times 10^{-16}$), which has been shown to have oncogenic functions through interactions with SMAD2, SMAD3 and SMAD4, key mediators of TGFβ signaling[33–35]. Our analysis also implicates rs9547700 (top GWAS variant rs12427846; $r^2 = 0.96$, $D' = 0.98$) as the functional basis of the 13q13.3 locus, and the risk allele is associated with reduced transcriptional activity and decreased *SMAD9*

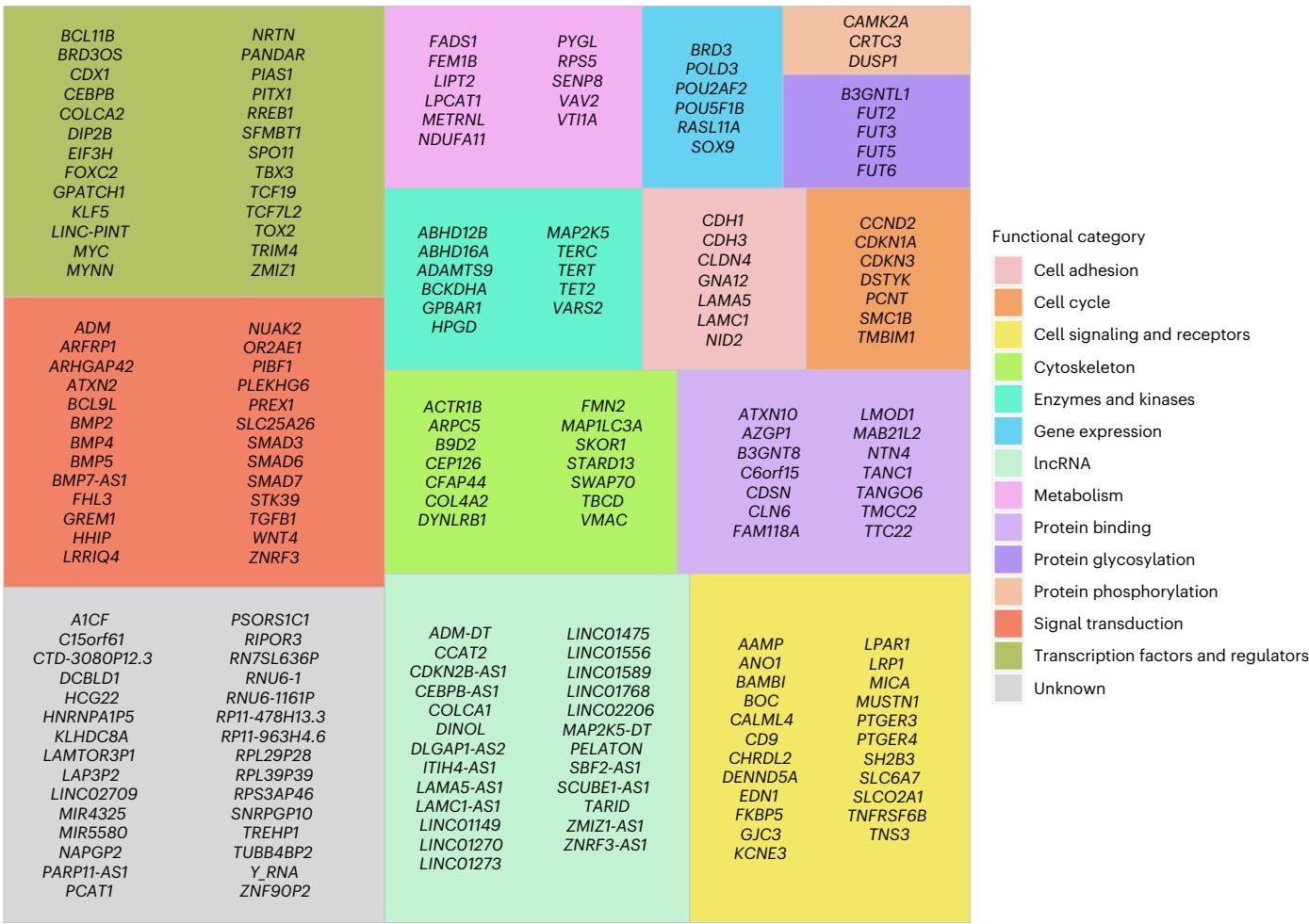

**Fig. 4 | Treemap of the candidate target genes, which are grouped by functional category.** Genes that were identified in the integrated analysis were classified according to their biological or cellular function. The size of the box is proportional to the number of genes in the category.

expression, further emphasizing the central role of genetic variation in TGFβ signaling pathways as a determinant of genetic susceptibility (Supplementary Fig. 6).

**Gene list analysis**

By performing this integrated analysis, we identified a set of 208 genes from the GWAS loci (Fig. 4 and Supplementary Table 6) and showed here a direct link between the risk variant and an implicated gene. To determine which of the target genes that we identified are already known to have a role in CRC, and more broadly cancer, we used the text mining tool OncoScore[36], which examines text from all available studies in the biomedical literature. To complement this analysis, we queried semantic predications within the Semantic MEDLINE Database[37] using MELODI Presto[38]. An integration of the results from these searches revealed that 142 of the 208 candidate target genes that we identified appear to have no documented role in CRC, and 47 of these presently have no established role in any cancer (Supplementary Tables 7 and 8).

One of the aspirations of GWASs is to inform therapeutics. To investigate the potential clinical utility of the CRC target genes identified at risk loci, we used oncoEnrichR[39] to explore multiple sources of functional and drug curation, including Open Targets[40,41] and DepMap[42]. For ten of the genes, there are already approved drugs that provide an opportunity for repurposing (Supplementary Table 9). These include crofelemer and misoprostol. Crofelemer inhibits ANO1, a calcium-activated chloride channel, which has a role in epithelial fluid secretion, and the gene is overexpressed in CRC. Misoprostol

is a PTGER3 (prostaglandin receptor) agonist, potentially capable of addressing the downregulation of this receptor in tumors. In addition to these, several of the target genes identified are attractive drug targets, with 44 having clinical or discovery precedence, and a further 31 are likely to be tractable (Supplementary Table 10). Based on CRISPR knockout data, genomic biomarkers and patient data[43], *TBCD*, *KLF5* and *SOX9* are also predicted to be promising therapeutic targets in CRC, as are *CCND1*, *CDH1*, *MYC* and *POU5F1B* in many different types of cancer (Supplementary Table 11).

After investigating regulatory networks in the gene list, we identified transcription factor regulatory interactions in cancer and normal cells. It was possible to observe sets of 'hub' genes, including *MYC*, *MYNN*, *EGR1*, *ZNF263*, *CTCF* and *SP1* (Supplementary Fig. 7). Formally testing for molecular pathways enriched in the target genes revealed that the genes were enriched in TGFβ-related pathways (TGFβ signaling pathway, Kyoto Encyclopedia of Genes and Genomes (KEGG), $P_{\text{adjusted}}$ ($P_{\text{adj}}$) $= 4.31 \times 10^{-6}$; TGFβ signaling activation by blocking of tumor suppressors, Elsevier Pathway Collection, $P_{\text{adj}} = 4.4 \times 10^{-5}$; Hippo signaling pathway, KEGG, $P_{\text{adj}} = 6.50 \times 10^{-5}$; Wnt signaling pathway, KEGG, $P_{\text{adj}} = 9.38 \times 10^{-3}$), as well as in cancer-related pathways (pathways in cancer, KEGG, $P_{\text{adj}} = 1.43 \times 10^{-6}$). Given the central role of these pathways in CRC development, these findings expand opportunities for indirect targeting; for example, the use of porcupine inhibitors to indirectly target Wnt pathway activity[44]. Hence, adapted forms or modified dosing regimens of these drugs may offer alternative treatment options.

## Discussion

To prioritize functional variants for the identification of CRC susceptibility genes at risk loci, we systematically scored multiple genetic and functional features as well as assayed allelic transcriptional activity. Integration of these data nominated 208 variants at 170 risk loci, few of which have previously been formally investigated.

Our data support tissue-specific transcriptional regulation as a major mechanism through which GWAS variants influence CRC risk. Although 24% (40 out of 170) of the loci had a single Tier 1 variant, 58% (98 out of 170) featured more than one equally plausible functional variant. The potential of multiple functional variants at some loci to underscore CRC risk and plausibly target more than one gene is consistent with a study reporting that multiple causal regulatory variants in high linkage disequilibrium are responsible for a subset of lymphoblastoid cell eQTLs[5]. In 48% of the GWAS risk loci, the candidate target gene was the closest to the gene or intronic, often localizing within the same TAD. This is in line with the Open Targets gold standard dataset[40], and this proximity effect has previously been noted and proposed to reflect evolutionary conservation[13]. For an appreciable proportion of risk loci, we found no obvious candidate genes, largely due to a paucity of functional data in these regions. This may be indicative of alternative mechanisms of action that were not explored here. For example, it has recently been proposed that the mechanistic basis of the 8q23.3 risk locus is a consequence of variable number tandem repeats[45].

We acknowledge that this study has some limitations. First, MPRA-significant variants were not identified for 36% of the GWAS loci. The functional basis of these risk loci might operate through mechanisms that cannot be tested by MPRAs. However, we cannot exclude technical issues or simply lack of statistical power to demonstrate a difference in allelic transcriptional activity. Second, for 96 loci, we could not assign a target gene using eQTLs. We have sought to address the cellular context of eQTLs, analyzing both normal and tumor data, although failure to demonstrate a relationship may reflect a lack of statistical power, especially for lower-frequency variants. Therefore, rather than rely solely on eQTLs, we performed an ABC-model-based analysis utilizing epigenomic features and Micro-C data to predict the enhancer–gene connections.

Accepting these caveats, we performed a multilayered analysis that enabled us to nominate the probable causal variants for the CRC risk loci and implicate 208 gene targets as the biological basis of associations. Only six of the genes we identified (*BCL9L*, *CDH1*, *SMAD3*, *SOX9*, *TBX3* and *TCF7L2*) are established CRC driver genes[46] (that is, genes with recurrent nonsynonymous somatic mutations in CRC under positive selection). This suggests a model by which genetic predisposition indirectly affects oncogenesis. In addition to emphasizing the role of genetic variation in established CRC genes and pathways, we identify candidate target genes with hitherto no previously well-established role. Notably, these include components of the calmodulin superfamily, *CALML4* and *CAMK2A*. The calmodulin pathway is the principal calcium sensor regulating a myriad of vital biological processes, including cell proliferation, programmed cell death and autophagy, and is increasingly viewed as an attractive therapeutic target[47]. *SLCO2A1*, which has a role in the synthesis and clearance of prostaglandins, along with *FADS1*, also highlights the importance of inflammation and the immune response in CRC development. The identification of *ATXN10* and *ATXN2* as candidate target genes provides support for the involvement of the Ras–MAP kinase pathway and EGFR trafficking in CRC development. *BCKDHA* catalyzes the breakdown of branched-chain amino acids, the dysregulation of which is recognized to have a role in the progression of a range of cancers[48].

In summary, we provide further insight into the functional basis of risk loci, implicating novel genes in the development of CRC, which expands the potential for therapeutic targeting. Our analysis provides an outline for a generalized strategy to profile disease-associated GWAS loci using high-throughput variant screening in concert with multilayered functional annotation.

## Online content

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

## Methods

### Ethics

For the eQTL data, all participants provided informed written consent, and the research was approved by local research ethics committees (SOCCS 11/SS/0109 and 01/0/05) and National Health Service management (SOCCS 2013/0014, 2003/W/GEN/05).

### GWAS statistics and definition of risk loci

GWAS summary association statistics were obtained from the recently published GWAS meta-analysis of 100,204 CRC cases and 154,587 controls[2]. Risk loci were defined as variants with $P < 5 \times 10^{-8}$ and that were at least 500 kb apart. To identify secondary signals inside this window, a conditional analysis was performed on the meta-analysis summary statistics using genome-wide complex trait analysis with conditional and joint analysis[49]. As the GWAS data were based on east Asian and European individuals, we used genotyping data from 6,684 unrelated individuals of east Asian ancestry and 4,284 individuals of European ancestry from the UK10K project[50] and the 1000 Genomes Project[51], respectively, as a reference for an estimation of linkage disequilibrium. The conditional analysis was performed on each population separately, and the data were combined using a meta-analysis, retaining associations where $P_{conditional} < 5 \times 10^{-8}$. In total, there were 204 autosomal variants identified, which mapped to 170 loci.

### Cell lines and cell culture

CRC cell lines were cultured in 5% $CO_2$ at 37 °C, with SW403 (ACC294, DSMZ), SW480 (ACC313, DSMZ) and SW948 (91030714, ECACC) grown in DMEM (Gibco), HT29 (ACC299, DSMZ) in McCoy's 5A (Modified) Medium (Gibco) with GlutaMAX Supplement (Gibco), CL11 (ACC467, DSMZ) in DMEM/F-12 (Gibco), and C32 (ECACC) in Iscove's Modified Dulbecco's Medium (Gibco). Media were supplemented with 10% (20% for CL11) heat-inactivated FBS (Sigma). The normal colon crypt cell line HCEC-1CT (CkHT039-0229, Evercyte) was cultured in a 4:1 ratio of DMEM and Medium 199 (Gibco) supplemented with ColoUp medium (Evercyte) at 37 °C, with 3% $O_2$ and 5% $CO_2$. All cell lines were cultured until they reached 90% confluency and then passaged using TrypLE (Gibco).

### MPRAs

**Variant selection.** We used MPRAs to identify variants exhibiting transcriptional differences. The nature of the assay requires that the variants to be tested are predefined, and the number of variants tested were constrained by oligonucleotide synthesis chip capacity. In light of this, using data from the CRC GWAS, we selected variants for MPRA testing by first considering all variants in a 500-kb window spanning each primary or conditional association (that is, ±250 kb around each lead variant) whose $P$ values were within three orders of magnitude of that of the lead variant. As this might not capture functional variants that remain highly significant at some loci (that is, where the lead variant has an extremely strong association), we also considered variants having $-\log_{10}(P_{variant}) > 0.7 \times (-\log_{10}(P_{lead\ variant}))$, stipulating an $r^2$ of >0.2 for the lead variant and a $P_{variant}$ of $<10^{-5}$ in the GWAS. A total of 100 control variants were also evaluated: 50 were derived from common variants (minor allele frequency > 0.05) that mapped to repressive regions (greater than fourfold enriched versus input) as defined by the NIH Roadmap Epigenomics Project colonic (E-075) H3K27me3 mark, and the remaining 50 were randomly generated.

**Variant oligonucleotide library design.** For each variant, 100-bp flanking sequences were added, yielding genomic probe sequences of 201 bp (100 + 1 + 100). Oligonucleotides containing an SceI restriction site, which was used for cloning, were excluded. During library synthesis, probe orientation was determined by the addition of two adapter sequences (AGGACCGGATCAACT and CATTGCGTGAACCGA) at either the 5′ and 3′ ends or the 3′ and 5′ ends relative to the probe sequence.

Each variant had four probes: one for each combination of forward and reverse strands and one for each reference and alternative allele. Library synthesis was performed by Twist Bioscience.

**Library construction, transfection and sequencing.** A lentiviral MPRA was carried out as previously described[52]. In brief, the MPRA library was amplified using 12 cycles of PCR, with adapter sequences as primers. All PCR reactions were performed using Q5 High-Fidelity 2X Master Mix (NEB). Subsequent rounds of PCR incorporated a random 15-base polymer barcode sequence for probe identification. Barcoded probes were incorporated into a pLS-SceI vector (a gift from N. Ahituv; Addgene plasmid no. 137725) by Gibson assembly using NEBuilder HiFi DNA Assembly Cloning Kit (NEB). After ligation, 100 ng of plasmid was transfected into NEB Stable Competent *E. coli* (High Efficiency) (NEB) using an Eppendorf Eporator at 1.8 kV. Bacteria were plated on carbenicillin (500 μg ml⁻¹) agar plates. A total of $2 \times 10^6$ colonies, sufficient for 100 unique barcodes per probe, were collected, and plasmid DNA was purified using ZymoPURE II Plasmid Maxiprep Kits (Zymo Research) before Illumina-based library preparation. Barcode-to-probe association was carried out by sequencing 4 nM of the pLS-SceI library on an Illumina MiSeq using a MiSeq Reagent Kit v2 (300 cycles) with three custom primers. Primer sequences are provided in Supplementary Table 9. Custom primers were diluted to a final concentration of 0.5 μM and added as follows: pLSmP-ass-seq-R1 (forward probe) read 1 (146 cycles), pLSmP-ass-seq-R2 (reverse probe) read 2 (146 cycles) and pLSmP-ass-seq-ind1 (forward barcode) index read 1 (15 cycles). The sample index read 2 (10 cycles) was performed using the default Illumina P5 primer.

Lentivirus particles were produced in HEK239T cells (CRL-11268, ATCC). For one T175 flask, 10 μg pLS-SceI, 6.5 μg psPAX2 and 3.5 μg pMD2.G were diluted in 2 ml of Opti-MEM (Gibco) and 40 μl of Turbo-Fect (Thermo Fisher Scientific) and added according to the manufacturer's guidelines. Other virus preparation steps were carried out as previously described[52]. HT29, SW403 and HCEC-1CT cells were used for enhancer quantitation. Cells were transduced with a viral moiety of infection of 80 based on cell-line-specific or batch-specific viral transduction efficiencies using 8 μM polybrene (Sigma). SW403 cells were transduced before attachment (reverse transduction), and the other cell lines were allowed at least 24 h to attach. After 24 h, the medium was removed, and the cells were incubated for an additional 48 h. The cells were lysed, and the RNA and DNA were purified using an AllPrep DNA/RNA Kit (Qiagen). DNA and RNA library preparation and sequencing were performed as previously described[52]. DNA and RNA samples were uniquely indexed, and a 16-bp random molecular identifier was added using PCR to eliminate optical duplicates. For each cell line, three DNA and three RNA replicate libraries were combined in equimolar amounts. DNA and RNA libraries from each cell line were mixed at a 1:3 ratio and diluted to 7.89 nM for sequencing. MPRA libraries were sequenced using a NovaSeq 6000 (Illumina) using the following primers: pLSmP-ass-seq-ind1 (forward barcode) read 1 (15 cycles), pLSmP-bc-seq (reverse barcode) read 2 (15 cycles) and pLSmP-UMI-seq (forward unique molecular identifier) index read 1 (16 cycles) (Supplementary Table 12). Sample indexes (index read 2, 10 cycles) were sequenced using the default Illumina P5 primer.

**Data analyses.** Raw sequencing data were converted to FASTQ format using bcl2fastq (Illumina). The MPRAflow[52] pipeline v2.3.5 was used to associate and count the number of barcodes associated with each probe sequence. To identify the different alleles for each variant, the FASTQ files were modified to include the forward library adapter (AGGACCGGATCAACT). This sequence was also added to the design FASTA file used by MPRAflow for alignment. For a sequence to be associated with a given barcode, it had to be a perfect match to the library sequence, which was enforced using a CIGAR string of 230M. For statistical analysis of the MPRA data, we used MPRAnalyze[53] v1.12.0, which uses a

nested pair of generalized linear models designed to estimate noise in the DNA and RNA libraries. We filtered the barcodes and collected those that contained all four allele-specific probes (that is, fwd_ref, fwd_alt, rev_ref and rev_alt), and we only retained a barcode if there was a DNA read present with a corresponding RNA read in the same replicate. Library size correction factors were estimated according to the replicate number, allele type (alternative or reference) and stand (forward or reverse) using the upper quantile of nonzero values for depth estimation. Owing to the large number of barcodes, we used the 'scaled' option, which uses the DNA counts directly as estimates rather than generating a DNA model. There was a strong correlation between the scaled analysis and the full model in a downsampled dataset (Supplementary Fig. 8). A likelihood test was performed to test the effect of the allele using the direction and replicate as covariates.

## ChIPmentation
ChIPmentation was performed on histone marks H3K4me1 (C15410194, Diagenode), H3K4me3 (C15410003-50, Diagenode), H3K27ac (C15410196, Diagenode), H3K27me3 (C15410195, Diagenode), H3K36me3 (C15410192, Diagenode) and CTCF (C15410210-50, Diagenode) for the C32, CL11, HT29, SW403, SW480, SW948 cell lines using a published protocol[54], with minor modifications, as detailed in the Supplementary Note. Data processing was performed using the Nextflow nf-core chipseq pipeline v1.2.1 (ref. [55]) with default parameters.

## Omni-ATAC
ATAC-seq was performed on the C32, CACO2, CL11, HT29, SW403, SW480, SW948 and HCEC-1CT cell lines as previously described[56]. Experimental protocols are detailed in the Supplementary Note. Data processing was performed using the Nextflow nf-core atacseq pipeline v1.2.1 (ref. [57]) with default parameters. Peaks for both the ChIP–seq and ATAC-seq data were consistent across all cell lines (Supplementary Fig. 9).

## Micro-C
We generated Micro-C chromatin interaction maps of the CL11, HT29, SW403, SW480 and SW948 cell lines as previously described[58, 59]. Experimental protocols are detailed in the Supplementary Note. The data were analyzed using JuicerTools v1.22 (ref. [60]) to count valid interactions. We required valid interactions of >90% for classification as *cis*-contacts, of which 60–70% had to be short-range *cis*-contacts. If the metrics were satisfactory, the pooled library was sequenced on a NovaSeq 6000 (Illumina) to a depth of at least 300 million reads per library, using 100 bp paired-end sequencing.

We used the nf-distiller pipeline[61] v0.3.4 to generate the interaction maps from the raw FASTQ files, using matrix balancing normalization and binning at 1 kb. FitHiC2 (ref. [62]) was used to call significant interactions, merging adjoining bins with significant interactions. TADs and compartments were identified using cooltools[63] v0.5.4 with 30-kb and 100-kb windows, respectively, and binning at 10 kb. Compartments were determined using an eigendecomposition of the contact matrix. The GC content of each bin was used as a phasing track. Active and inactive compartments are defined as having a positive and negative value for the first eigenvector, respectively.

## RNA extraction and library sequencing
RNA sequencing of the C32, CL11, HT29, SW403, SW948 and HCEC-1CT cell lines was performed. The experimental protocols are detailed in the Supplementary Note. Analysis of the RNA-seq data was performed using the RNAflow pipeline[64] v1.4.1 with default parameters.

## Cell-type specificity of risk variants
To identify the cell types through which CRC risk variants exert their effects, we analyzed single-cell gene-expression profiles across different tissues using the Tabula Sapiens v4 dataset[23] (~500,000 cells from

24 organs from 15 normal human subjects) and across different intestinal regions using the Gut Cell Atlas[24] (~125,000 cells from 86 healthy adults from 11 distinct locations in the gut). We used scDRSs[65] v1.0.1 to link the scRNA-seq data with polygenic disease risk at a single-cell resolution, independent of cell type. In brief, using the CRC GWAS association summary statistics, MAGMA[66] v1.10 defined a putative set of disease genes. Using the top 1,000 putative genes, a disease score was calculated as a function of the GWAS z-scores and the scRNA-seq expression values. Cell-specific association *P* values were calculated by comparing normalized disease scores to an empirical distribution of normalized scores across all control gene sets and all cells.

## Histone mark enrichment analysis
To examine enrichment in specific histone marks across the risk loci, we adapted the variant set enrichment method described previously[67,68]. In brief, for each risk locus, a region of strong linkage disequilibrium (defined as $r^2 \geq 0.8$ and $D' \geq 0.8$) was determined, and variants mapping to these regions were termed the associated variant set (AVS). ChIP–seq data for the H3K4me3, H3K27ac, H3K4me1, H3K27me3, H3K9ac, H3K9me3 and H3K36me3 chromatin marks from up to 128 cell types were obtained from the NIH Roadmap Epigenomics Project data[25]. For each mark, the overlap of the positions of variants in the AVS and the ChIP–seq peaks was determined to produce a mapping tally. A null distribution was generated by randomly selecting variants with the same linkage disequilibrium characteristics as the risk-associated variants, and a null mapping tally was calculated. This process was repeated 50,000 times, and approximate *P* values were calculated as the proportion of permutations for which the null mapping tally was greater or equal to the AVS mapping tally.

## ChromHMM
We used ChromHMM[69] v1.24 to predict chromatin states using the H3K4me1, H3K4me3, H3K27ac, H3K27me3 and H3K36me3 histone marks. The BAM files from the nf-core chipseq pipeline described above were binarized, and a 15-state model was predicted (Supplementary Fig. 10). States were annotated using previously published annotations[69–71].

## Fine-mapping of risk loci
Using summary data from the CRC GWAS, we defined flanking regions 500 kb on either side of the most significant variant at each risk locus. We performed statistical fine-mapping of these CRC risk loci using PolyFun[26] v1.4.1 and susieR[27] v0.11.92. We calculated the previous causal probabilities nonparametrically using the established PolyFun protocol, which estimates the per-SNP heritability for each variant, weighted by the functional annotations. Annotation data were gathered from the baseline-LF v2.2 annotation data[26,72] provided by the A. Price group (https://alkesgroup.broadinstitute.org/LDSCORE) using the CRC-specific ChIP–seq and open chromatin data that were generated in-house (Supplementary Table 1). Linkage disequilibrium scores were calculated using data from 45,498 disease-free European individuals in the Genomics England dataset (https://re-docs.genomicsengland.co.uk/aggv2). Using the probabilities estimated by PolyFun, we fine-mapped loci across a 500-kb window using the Sum of Single Effects model, which was implemented in susieR. For loci with one independent variant, we set the maximum number of causal variants to two, as susieR is unable to use linkage disequilibrium information for a single variant. For loci with multiple independent variants, we performed fine-mapping of the region including all independent variants, and set the maximum number of causal variants equal to the number of independent variants. The output from susieR included a posterior inclusion probability (PIP) for each variant and the 95% credible set that the variant belongs to. Variants with PIPs of >0.001 and that cumulatively reached a probability of 0.95 were included in a credible set.

## Transcription factor binding

We used TOBIAS[73] v0.14.0 to predict transcription factor binding using the BINDetect method. Using the ATAC-seq data from the C32, CACO2, CL11, HT29, SW403, SW480, SW948 and HCEC-1CT cell lines in conjunction with the JASPAR 2022 core nonredundant transcription factor motif database[28], which was filtered to motifs found in humans, we predicted whether there were any potential transcription factors bound in open chromatin. The TOBIAS scores indicate how well the transcription factor motif matches the genomic sequence.

We performed an enrichment analysis of bound transcription factors according to the NIH Roadmap Epigenomics Project histone analysis. The number of transcription factors predicted to bind to each of the GWAS regions (based on the selected variants) were counted. A null distribution was generated by randomly selecting variants with $P_{variant} > 0.95$; a window of a size equivalent to that of the GWAS data was formed, and a null transcription factor count was calculated. This process was repeated 50,000 times, and approximate $P$ values were calculated as the proportion of permutations for which the null transcription factor count was greater or equal to the GWAS transcription factor count.

## ABC model for prediction of enhancer–gene interactions

To predict enhancer–gene connections in each cell line, we used ABC[22] v0.2.2 in conjunction with data from ATAC-seq, H3K27ac ChIP–seq, Micro-C and RNA-seq. The analysis was performed as previously described[22] using default parameters. In brief, we investigated the 150,000 gene–enhancer interactions with the highest ABC scores for all enhancer regions within 5 Mb of the TSS of a gene. Enhancer regions were filtered such that those overlapping the GWAS risk loci were retained.

## eQTLs and SMR

Comprehensive details about the RNA-seq and whole-genome sequencing data from the SOCCS and GTEx datasets are described in ref. 74. In brief, the GTEx data were derived from 367 postmortem transverse colon samples, and the SOCCS data were derived from the normal colon or rectum mucosa of 223 healthy individuals. eQTL analysis was performed using Matrix eQTL v2.3 (ref. 75) on probabilistic estimation of expression residuals-adjusted residuals[76], and age, sex, batch and a number of hidden covariates equal to one quarter of the sample size in both datasets were taken into account. The variants tested were limited to those within 0.5 Mb of lead variants, with a minor allele frequency of >0.01 and associations with genes within 1 Mb. Per-dataset results underwent a meta-analysis using a fixed-effects inverse variance-weighted model in META[77] v1.7. SMR[32] v1.3.1 analysis was performed using the eQTL results from the meta-analysis and GWAS summary statistics from ref. 2 using default parameters. As SMR is performed only on the top eQTLs for each gene, $P_{SMR}$ values were Bonferroni-corrected for multiple testing based on the number of genes within each risk locus. We retained results with $P_{SMR-corrected} < 0.05$ and $P_{HEIDI} > 0.05$. To analyze the preservation of CRC risk-associated eQTL effects in tumors, eQTL summary statistics from TCGA COAD ($n = 286$) and READ ($n = 94$) samples were obtained from PancanQTL[78], underwent a meta-analysis and subjected to SMR as described above.

## 3D chromatin structure disruption

To predict the effect of variants on the 3D structure of DNA, we used Akita[31] v0.6, which uses a deep learning framework. The Micro-C data were binned into 1,024 ($2^{10}$)-bp sets. Data were preprocessed using default parameters, except for a sequence length of 1,048,576 ($2^{20}$) bp and a crop length of 65,536 ($2^{16}$) bp. The model was trained using default parameters, with 10% of the data used for testing and 10% used for validation. We performed in silico mutagenesis on a nucleotide level on 200-bp regions centered on each tested variant. Disruption scores were calculated as the L2 norm of the predicted differences between the contact maps for each allele.

## Scoring of variants

To prioritize the variants in each of the risk loci, we adopted the following scoring scheme:

- MPRA: variants with an FDR of ≤$10^{-3}$ were given a score of 2, and those with an FDR of ≤0.05 were given a score of 1. Each cell line was considered separately.
- Statistical fine-mapping: variants with a PIP of >0 were given a score of 1 (that is, the variant was part of a credible set), and those with a PIP of >0.5 were given a score of 2.
- Chromatin annotation: based on ChromHMM annotation, variants that fell within either a promoter or an enhancer region were given a score of 2, and those that fell within regions with weak predicted states (that is, with lower emission parameters) were given a score of 1.
- Open chromatin: if the variants fell within an ATAC-seq peak, then they were given a score of 2.
- SMR: for both the normal and tumor samples, if a variant was associated with a gene identified using SMR, then it was given a score of 2.
- Akita: if >25% of the variants within 100bp of the tested variant had a disruption score in the top 10% of all disruption scores, then it was given a score of 2. If >25% of the variants within 100bp of the tested variant had a disruption score in the top 20% of all disruption scores, then it was given a score of 1.
- CTCF: if the variants fell within a CTCF peak, then they were given a score of 2.
- Long-range interaction: using the output from FitHiC (filtered using $-\log_{10}(P) \geq 2$), if the variant fell within one end of a Micro-C contact and the other end was within a gene body, then it was given a score of 1. If the other end of the interaction contained a TSS of a gene, then the variant was given a score of 2.

As many of these analyses were performed on multiple cell lines, it was necessary to find a scoring consensus across cell lines for collation of the scores. For the ATAC-seq, CTCF, Micro-C and ChromHMM data, this consensus was that the annotation had to be present in >50% of the cell lines. For the ABC model and the transcription factor binding prediction, we performed a binomial analysis of the scores from the respective analysis tools. For each analysis, we identified the number of times that the score was in the 90th percentile. We calculated the probability of the occurrence of this score using the binomial distribution survival function. If $P < 0.05$, the tested variant was assigned a score of 1, and if $P < 0.01$, it was given a score of 2.

The annotation scores for each variant were summed, and the scores were ranked. The variants with scores in the top 20% of all scores were designated as Tier 1 variants, those with scores in the bottom 50% as Tier 3 and the remainder as Tier 2.

## Gene prioritization

To link the variants with genes, we used the data from the Micro-C TSSs, the ABC model and SMR of the tumor and normal samples. We focused on the Tier 1 variants, as they provided the most information regarding annotations. A gene annotation had to be present in at least two of the annotation sources to be suggested as a putative target gene. If no genes reached this threshold, we included the interactions related to the Micro-C data within a gene body and included these as weak predictions. For any genes that did not have any Tier 1 variants, we used the Tier 2 variants instead, and any genes that were found were labeled as weak predictions. Finally, if no genes were identified throughout the process and the variant was intronic to a gene, that gene was used as a weak prediction.

## Gene evidence

To formally examine whether target genes were known to be associated with cancer (and specifically CRC), we used OncoScore[36] v1.30.0,

a text mining tool that ranks genes by their association with cancer based on the available biomedical literature. We used an OncoScore of 21.09 as the threshold to define novelty. To complement this analysis, we also performed a literature survey in MELODI Presto (accessed 3 April 2024)[38] using semantic predications in the Semantic MEDLINE Database[37], which is based on all citations in PubMed. Within the Semantic MEDLINE Database, pairs of terms were connected by a predicate, which are collectively known as 'literature triples' (that is, 'subject term'–predicate–'object term'). We performed the analysis using the gene list as the subject and 'colorectal cancer' as the object. Driver genes were determined using intOGen (released 1 February 2020)[46] and restricted to those from colorectal cohorts (COADREAD). Gene distance information was obtained from HaploReg v4.1.

We used oncoEnrichR[39] v1.4.2.1 to analyze the gene sets. This tool provides a suite of analyses, including cancer associations, drug associations, synthetic lethality, gene fitness and protein–protein interactions.

Regulatory interaction data were obtained from the DoRothEA and OmniPath resources[79,80]. These datasets contains a list of previously identified transcription factor–target interactions that are scored based on multiple lines of evidence (namely, literature-curated resources, ChIP–seq peaks, transcription factor binding site motifs and gene-expression-inferred interactions). Regulatory interactions were inferred using gene expression in tumor samples (from TCGA) or normal tissues (from GTex).

Cell viability and gene essentiality data were obtained from the Cancer Dependency Map (DepMap, 2020_Q2 release), which provides information on a systematic genome-scale CRISPR–Cas9 drop-out screen in 912 cancer cell lines[42]. We restricted the analysis to the CRC cell lines from primary tumors (that is, nonmetastatic; $n = 37$). To identify putative therapeutic targets, we used the results from the Project Score database (2021_Q2 release)[42,43] in DepMap. This generates target priority scores based on the integration of CRISPR knockout gene fitness effects with genomic biomarker and patient data (accounts for 30% of the score and is based on evidence of a genetic biomarker associated with a target dependency, as well as tumor prevalence), and cell line fitness effects (accounts for 70% of the score and is based on gene fitness, genes expressed and genes not homozygously deleted). All genes are assigned a target priority score between zero and 100 from lowest to highest priority. A threshold score of 40 was established based on scores calculated for targets with approved or preclinical cancer compounds.

Drug tractability information was based on data from the Open Targets Platform[41], and pathway enrichment was performed using Enrichr (released 8 June 2023)[81].

### Reporting summary

Further information on research design is available in the Nature Portfolio Reporting Summary linked to this article.

### Data availability

GWAS data are available from GWAS Catalog (accession no. GCST90129505). Cell line data have been deposited in the European Genome-phenome Archive under the following accessions: EGAD50000000596 (MPRA), EGAD50000000294 (Micro-C), EGAD50000000295 (ChIP–seq, all marks), EGAD50000000296 (ATAC-seq), EGAD50000000297 (RNA-seq). Annotation data for all the GWAS regions are available on the University of California, Santa Cruz (UCSC) Genome Browser (https://genome.ucsc.edu/s/philip.law%40icr.ac.uk/CRC%20GWAS%20annotation). Single cell RNA-seq data were obtained from the Gut Cell Atlas (https://www.gutcellatlas.org) and the Tabula Sapiens project (https://tabula-sapiens-portal.ds.czbiohub.org). Transcription factor binding was based on data from JASPAR (https://jaspar.genereg.net). Functional annotations for the fine-mapping were obtained from the A. Price group (https://alkesgroup.broadinstitute.org/LDSCORE). Histone marks in

different tissues were obtained from the NIH Roadmap Epigenomics Project (https://egg2.wustl.edu/roadmap/web_portal). eQTL data were obtained from PancanQTL (http://bioinfo.life.hust.edu.cn/PancanQTL) and GTEx (https://gtexportal.org). Literature mining was performed in MELODI Presto (https://melodi-presto.mrcieu.ac.uk) using data from the Semantic MEDLINE Database (https://lhncbc.nlm.nih.gov/ii/tools/SemRep_SemMedDB_SKR.html). Gene annotation data were obtained from OmniPath (https://omnipathdb.org), DoRothEA (https://saezlab.github.io/dorothea), DepMap (https://depmap.org) and Open Targets (https://www.opentargets.org), and analyzed in oncoEnrichR (https://oncotools.elixir.no). Source data are provided with this paper.

### Code availability

No custom code was generated. Publicly available code was used for all aspects of data processing and analysis and is cited in the appropriate section of the Methods.

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

## Acknowledgements

At the Institute of Cancer Research (ICR), this work was supported by Cancer Research UK (CRUK) (C1298/A25514 to R.S.H.), the Wellcome Trust (214388) and the ICR Genomics Facility, which performed the sequencing. In Edinburgh, this work was supported by funding from CRUK (DRCPGM/100012 and C348/A12076 to M.G.D.), as well as the Cancer Research UK Scotland Centre in Edinburgh (CTRQQR-2021/100006 to M.G.D.), which provided infrastructure and staffing. B.T.H. was supported by a CRUK Ph.D. studentship, which was supervised by S. Farrington, at the Edinburgh CRUK Cancer Research Centre. We thank M. Schubach and M. Kircher for assistance in adapting the MPRAflow pipeline. We also thank M. Went, A. Gunnell and A. Everall for technical and statistical input. Figure 1 was created using BioRender.com.

## Author contributions

P.J.L. and R.S.H. designed and planned the study. P.J.L., C.M. and B.T.H. performed bioinformatic analysis. J. Studd, J. Smith, J.V. and M.M. performed experiments and analyzed data. P.J.L. integrated analyses and interpreted data. P.J.L. and R.S.H. wrote the manuscript. R.S.H. and M.G.D. provided supervision. All authors have read and approved the final version of the manuscript.

## Competing interests

The authors declare no competing interests.

## Additional information

**Extended data** is available for this paper at https://doi.org/10.1038/s41588-024-01900-w.

**Correspondence and requests for materials** should be addressed to Richard S. Houlston.

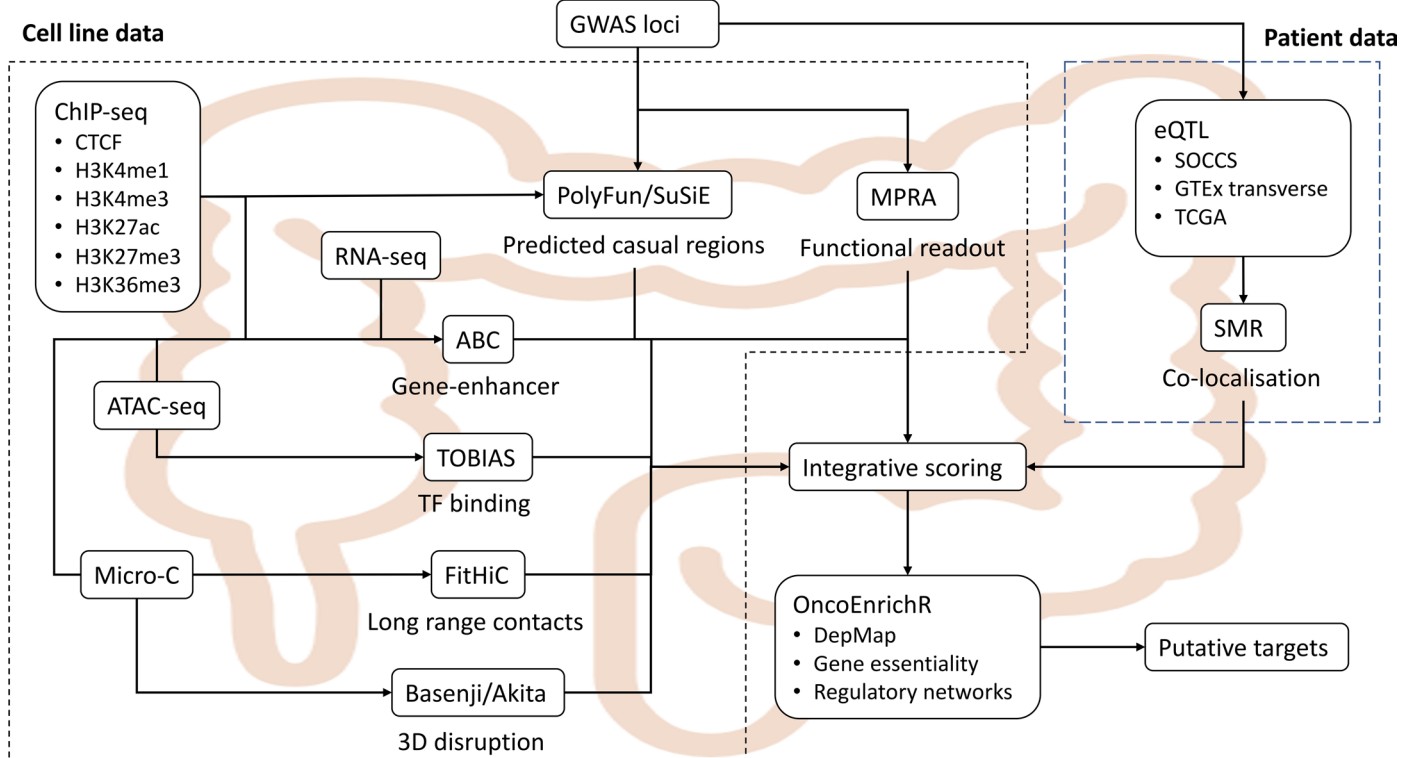

**Extended Data Fig. 1 | Detailed schematic of the analysis.** Detailed schematic of the analysis performed. Using the loci identified by the CRC GWAS, we annotated the regions using multiple functional modalities including massively parallel reporter assays (MPRA) to observe allelic effects on transcription, epigenetic marks (ChIP-seq), chromatin accessibility (ATAC-seq), gene expression (RNA-seq) and long-range chromatin interactions (Micro-C). ABC: Activity By Contact.

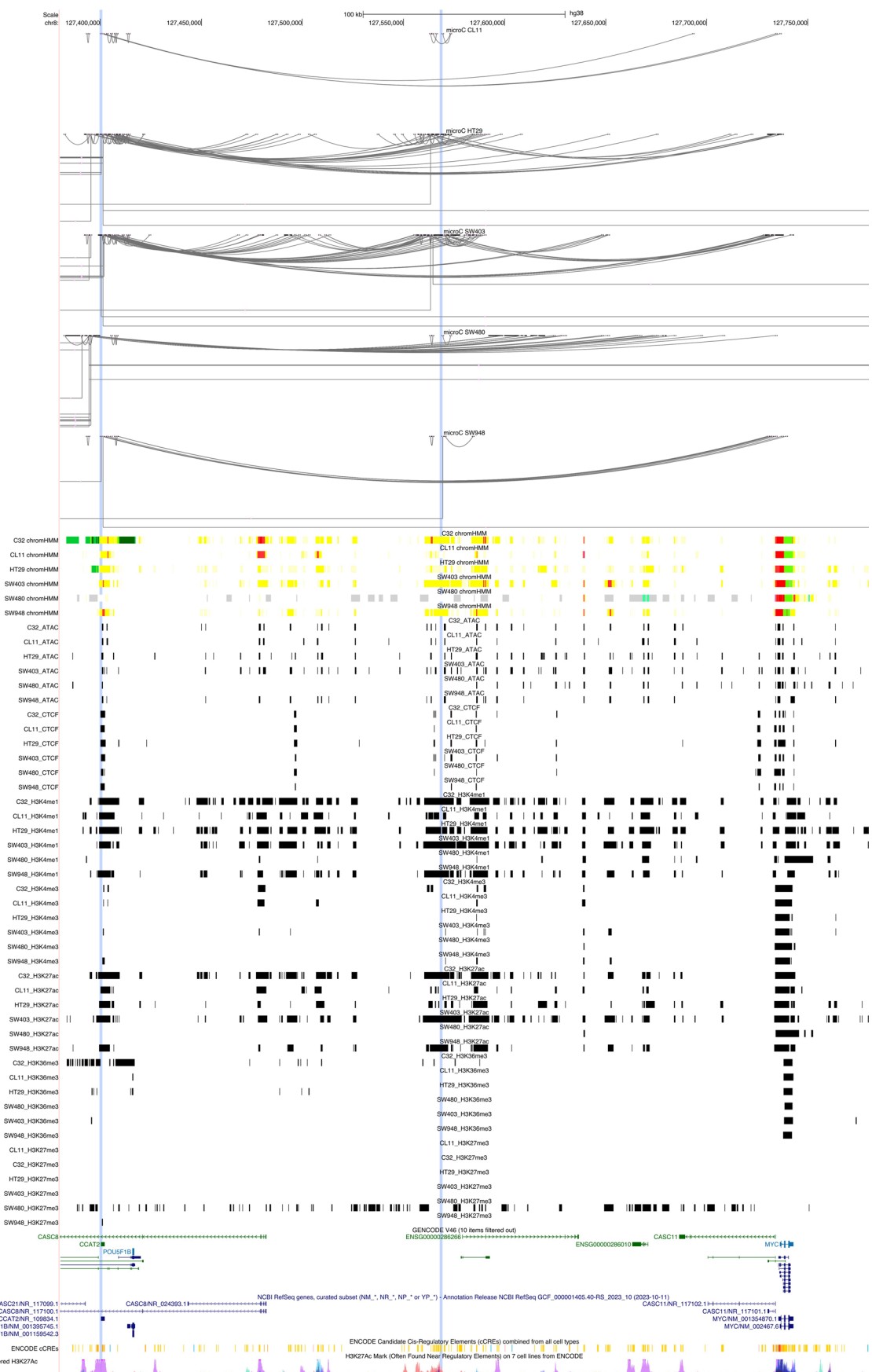

**Extended Data Fig. 2 | Detailed annotation for the variants in 8q24 locus.** Detailed functional annotation for the variants in 8q24 locus from UCSC Genome Browser, showing the Micro-C, chromHMM, ATAC-seq, and ChIP-seq data across the various cell lines. The putative variant, rs6983267, is highlighted in light blue (left). A secondary signal at rs4733767 is also shown (middle blue line).

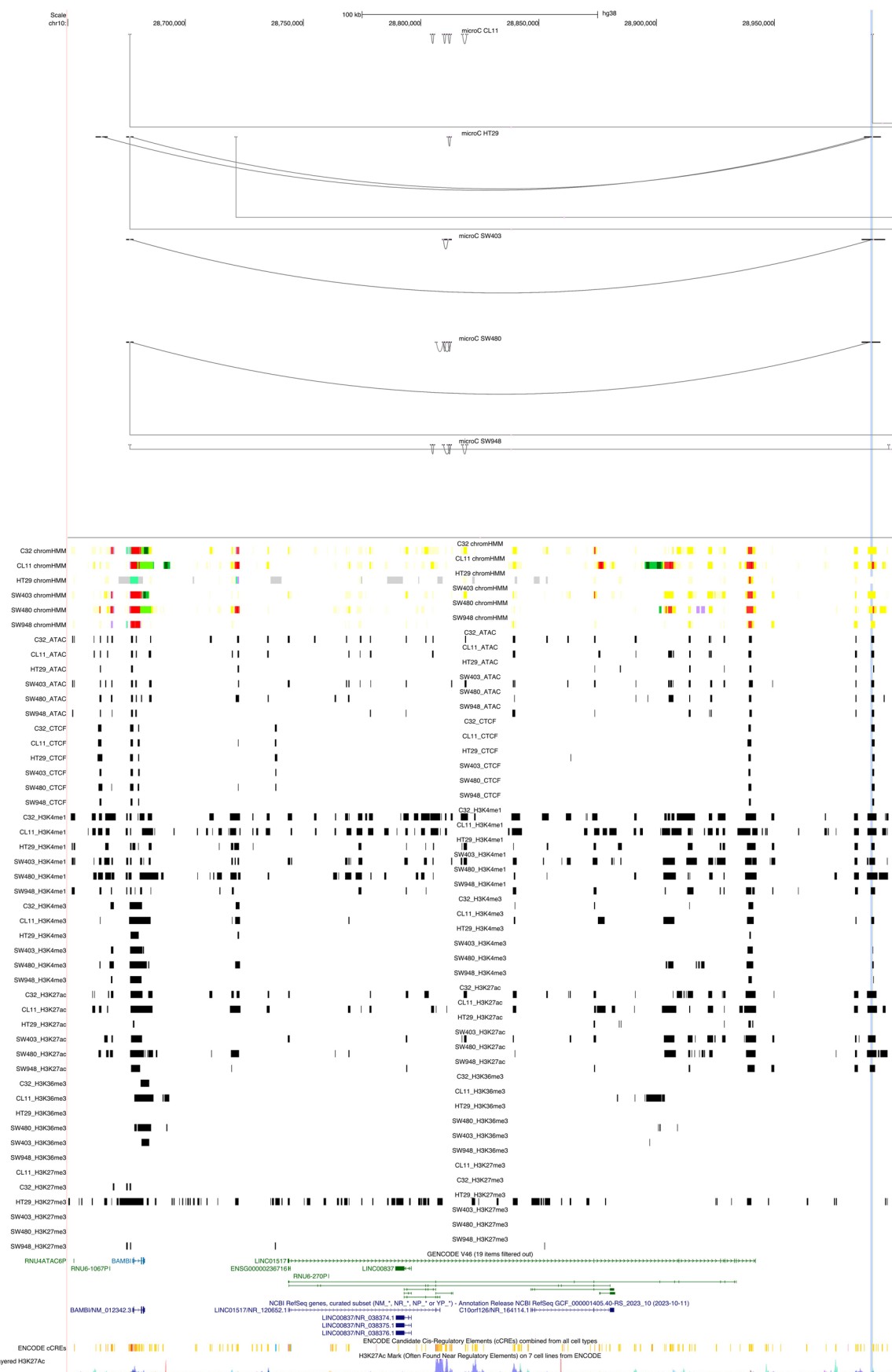

**Extended Data Fig. 3 | Detailed annotation for the variants in 10p12 locus.** Detailed annotation for the variants in 10p12 locus from UCSC Genome Browser. The putative variant, rs1248418, is highlighted in light blue.

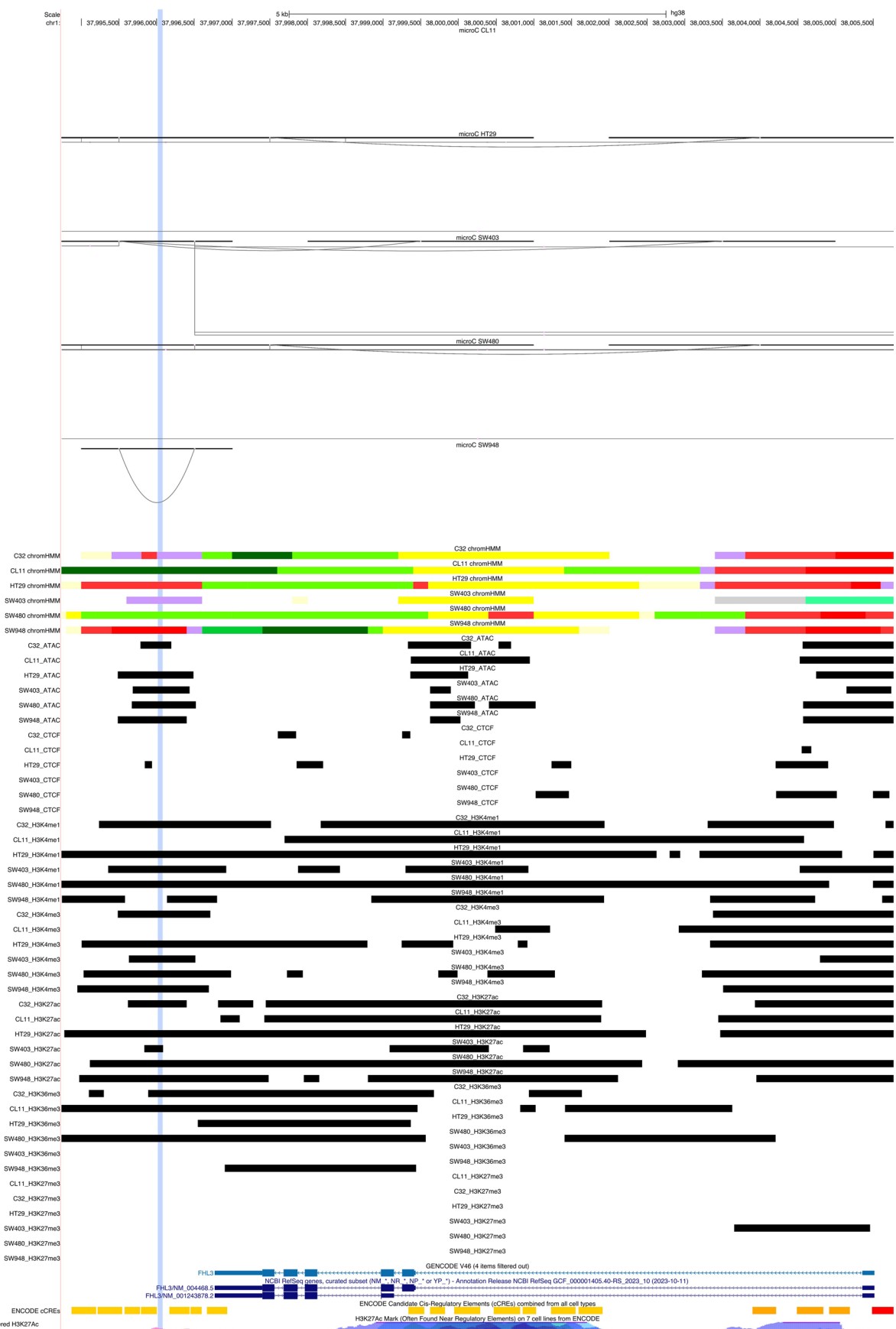

**Extended Data Fig. 4 | Detailed annotation for the variants in 1p34 locus.** Detailed annotation for the variants in 1p34 locus from UCSC Genome Browser. The putative variant, rs67631072, is highlighted in light blue.

# Reporting Summary

## Statistics

For all statistical analyses, confirm that the following items are present in the figure legend, table legend, main text, or Methods section.

| n/a | Confirmed | |
|---|---|---|
| ☐ | ☒ | The exact sample size (*n*) for each experimental group/condition, given as a discrete number and unit of measurement |
| ☒ | ☐ | A statement on whether measurements were taken from distinct samples or whether the same sample was measured repeatedly |
| ☐ | ☒ | The statistical test(s) used AND whether they are one- or two-sided *Only common tests should be described solely by name; describe more complex techniques in the Methods section.* |
| ☐ | ☒ | A description of all covariates tested |
| ☐ | ☒ | A description of any assumptions or corrections, such as tests of normality and adjustment for multiple comparisons |
| ☐ | ☒ | A full description of the statistical parameters including central tendency (e.g. means) or other basic estimates (e.g. regression coefficient) AND variation (e.g. standard deviation) or associated estimates of uncertainty (e.g. confidence intervals) |
| ☐ | ☒ | For null hypothesis testing, the test statistic (e.g. *F*, *t*, *r*) with confidence intervals, effect sizes, degrees of freedom and *P* value noted *Give P values as exact values whenever suitable.* |
| ☒ | ☐ | For Bayesian analysis, information on the choice of priors and Markov chain Monte Carlo settings |
| ☒ | ☐ | For hierarchical and complex designs, identification of the appropriate level for tests and full reporting of outcomes |
| ☐ | ☒ | Estimates of effect sizes (e.g. Cohen's *d*, Pearson's *r*), indicating how they were calculated |

*Our web collection on statistics for biologists contains articles on many of the points above.*

## Software and code

Policy information about availability of computer code

| Data collection | No software was used for data collection |
|---|---|
| Data analysis | ABC v0.2.2: https://github.com/broadinstitute/ABC-Enhancer-Gene-Prediction<br>Akita v0.6: https://github.com/calico/basenji<br>ChromHMM v1.24: http://compbio.mit.edu/ChromHMM/<br>cooltools v0.5.4: https://github.com/open2c/cooltools<br>distiller-nf v0.3.4: https://github.com/open2c/distiller-nf<br>Enrichr 2023.06.08 release: https://maayanlab.cloud/Enrichr<br>FitHiC2 v2.0.8: https://github.com/ay-lab/fithic<br>GCTA-COJO v1.92.3: https://yanglab.westlake.edu.cn/software/gcta/#COJO<br>HapoReg v4.1: https://pubs.broadinstitute.org/mammals/haploreg/haploreg.php<br>intOGen 2020.02.01 release: https://www.intogen.org/<br>JuicerTools v1.22: https://github.com/aidenlab/JuicerTools<br>MAGMA v1.10: https://cncr.nl/research/magma/<br>META v1.7 : https://mathgen.stats.ox.ac.uk/genetics_software/meta/meta.html<br>Matrix eQTL v2.3: https://www.bios.unc.edu/research/genomic_software/Matrix_eQTL/<br>MELODI Presto (accessed 2024.04.03): https://melodi-presto.mrcieu.ac.uk/<br>MPRAflow v2.3.5: https://github.com/shendurelab/MPRAflow<br>MPRAnalyze v1.12.0: https://www.bioconductor.org/packages/release/bioc/html/MPRAnalyze.html<br>nfcore-atacseq v1.2.1 https://nf-co.re/atacseq/<br>nfcore-chipseq v1.2.1 https://nf-co.re/chipseq |

OncoEnrichR v1.4.2.1: https://oncotools.elixir.no/
OncoScore v1.30.0: https://www.galseq.com/next-generation-sequencing/oncoscore-software/
PolyFun (downloaded 2023.09.13): https://github.com/omerwe/polyfun
RNAflow v1.4.1: https://github.com/hoelzer-lab/rnaflow
scDRS v1.0.1: https://github.com/martinjzhang/scDRS
susieR v0.11.92: https://github.com/stephenslab/susieR
SMR v1.3.1: https://yanglab.westlake.edu.cn/software/smr/#Overview
TOBIAS v0.14.0: https://github.com/loosolab/TOBIAS

For manuscripts utilizing custom algorithms or software that are central to the research but not yet described in published literature, software must be made available to editors and reviewers. We strongly encourage code deposition in a community repository (e.g. GitHub). See the Nature Portfolio guidelines for submitting code & software for further information.

# Data

Policy information about availability of data

All manuscripts must include a data availability statement. This statement should provide the following information, where applicable:
- Accession codes, unique identifiers, or web links for publicly available datasets
- A description of any restrictions on data availability
- For clinical datasets or third party data, please ensure that the statement adheres to our policy

GWAS data are available from GWAS Catalog (accession no. GCST90129505). Cell line data have been deposited the European Genome-phenome Archive (EGA) under the following accessions: EGAD50000000294 (Micro-C), EGAD50000000295 (ChIP-seq, all marks), EGAD50000000296 (ATAC-seq), EGAD50000000297 (RNA-seq), EGAD50000000596 (MPRA).  Annotation data for all the GWAS regions are available on UCSC Genome Browser (https://genome.ucsc.edu/s/philip.law%40icr.ac.uk/CRC%20GWAS%20annotation).
Single cell RNA-seq data were obtained from the Gut Cell Atlas (https://www.gutcellatlas.org/) and the Tabula Sapiens project (https://tabula-sapiens-portal.ds.czbiohub.org/). Transcription Factor binding was based on data from JASPAR (https://jaspar.genereg.net/). Functional annotations for the finemapping were obtained from the Alkes Price group (https://alkesgroup.broadinstitute.org/LDSCORE/). Histone marks in different tissues was obtained from the NIH Roadmap Epigenomics Project ( https://egg2.wustl.edu/roadmap/web_portal/). eQTL data were obtained from PancanQTL (http://bioinfo.life.hust.edu.cn/PancanQTL/) and GTEx (https://gtexportal.org/). Literature mining was performed using data from the Semantic MEDLINE Database (SemMedDB, https://lhncbc.nlm.nih.gov/ii/tools/SemRep_SemMedDB_SKR.html), as implemented in MELODI Presto (https://melodi-presto.mrcieu.ac.uk/). Gene annotation data were obtained from the OmniPath (https://omnipathdb.org/), DoRothEA (https://saezlab.github.io/dorothea/), DepMap (https://depmap.org/), and Open Targets (https://www.opentargets.org/), as implemented in oncoEnrichR (https://oncotools.elixir.no/).

# Research involving human participants, their data, or biological material

Policy information about studies with human participants or human data. See also policy information about sex, gender (identity/presentation), and sexual orientation and race, ethnicity and racism.

| | |
|---|---|
| Reporting on sex and gender | N/A |
| Reporting on race, ethnicity, or other socially relevant groupings | N/A |
| Population characteristics | N/A |
| Recruitment | N/A |
| Ethics oversight | N/A |

Note that full information on the approval of the study protocol must also be provided in the manuscript.

# Field-specific reporting

Please select the one below that is the best fit for your research. If you are not sure, read the appropriate sections before making your selection.

☒ Life sciences          ☐ Behavioural & social sciences          ☐ Ecological, evolutionary & environmental sciences

For a reference copy of the document with all sections, see nature.com/documents/nr-reporting-summary-flat.pdf

# Life sciences study design

All studies must disclose on these points even when the disclosure is negative.

| | |
|---|---|
| Sample size | Observational study in cell lines, so sample size determination was not necessary |
| Data exclusions | Standard data QC was performed on all assays. Sequencing quality was assessed to ensure sufficient quality. In the MPRA, we only retained reads that mapped perfectly to the barcode library, enforced using a CIGAR string of 230M. We additionally filtered the barcodes to those that contained all four allele specific probes (i.e. fwd_ref, fwd_alt, rev_ref, rev_alt), and additionally only |

retained a barcode if there was a DNA read present with a corresponding RNA read in the same replicate.
For the ChIP-seq, ATAC-seq, RNA-seq, and Micro-C reads were removed if:
 -reads mapped to blacklisted regions
 -reads were marked as duplicates
 -reads weren't marked as primary alignments
 -reads were unmapped
 -reads mapped to multiple locations
 -reads contained > 4 mismatches
 -reads had an insert size > 2kb
 -reads mapped to different chromosomes
 -reads arent in FR orientation
 -reads where only one read of the pair fails the above criteria

**Replication**

For the ChIP-seq and ATAC-seq, each assay was performed with two replicates. For the RNA-seq, each cell line was performed with three replicates. For the Micro-C, each cell line was performed with eight sub-libraries. For the MPRA, each cell line was performed with three replicates. All replicates succeeded.

**Randomization**

Observational study, so randomisation was not necessary

**Blinding**

Observational study, so blinding was not necessary

# Reporting for specific materials, systems and methods

We require information from authors about some types of materials, experimental systems and methods used in many studies. Here, indicate whether each material, system or method listed is relevant to your study. If you are not sure if a list item applies to your research, read the appropriate section before selecting a response.

## Materials & experimental systems

| n/a | Involved in the study |
|---|---|
| ☐ | ☒ Antibodies |
| ☐ | ☒ Eukaryotic cell lines |
| ☒ | ☐ Palaeontology and archaeology |
| ☒ | ☐ Animals and other organisms |
| ☒ | ☐ Clinical data |
| ☒ | ☐ Dual use research of concern |
| ☒ | ☐ Plants |

## Methods

| n/a | Involved in the study |
|---|---|
| ☐ | ☒ ChIP-seq |
| ☒ | ☐ Flow cytometry |
| ☒ | ☐ MRI-based neuroimaging |

## Antibodies

**Antibodies used**

All antibodies obtained from Diagenode. 5ug of target antibody was added per 3-5 x 10^5 cell lysate.
\# Catalog Antibody lot
1.C15410196 H3K27ac Antibody - ChIP-seq Grade A1723-0041D
2.C15410195 H3K27me3 Antibody - ChIP-seq Grade A0824D
3.C15410194 H3K4me1 Antibody - ChIP-seq Grade A1862D
4.C15410192 H3K36me3 Antibody - ChIP-seq Grade A1845P
5.C15410003-50 H3K4me3 Antibody - ChIP-seq Grade A8034D
6. C15410210-50 CTCF Antibody - ChIP-seq Grade A2354-0010

**Validation**

All chip-seq grade antibodies were validated by Diagenode (Hologic, USA) and the details of validation experiments are provided in the links. For example H3k27ac (C15410196) validation was performed in HeLa cell lines by quantitative PCR using primer pairs for active promoters of EIF4A2 and ACTB as positive controls while TSH2B and MYT1 promoters were used as negative controls. Further details are available for each antibody using the relevant links.

https://www.diagenode.com/en/p/h3k27ac-polyclonal-antibody-premium-50-mg-18-ml; H3K27ac Antibody (Diagenode Cat# C15410196 Lot# A1723-0041D)
https://www.diagenode.com/en/p/h3k27me3-polyclonal-antibody-premium-50-mg-27-ml; H3K27me3 Antibody (Diagenode Cat# C15410195 Lot# A0824D)
https://www.diagenode.com/en/p/h3k4me1-polyclonal-antibody-premium-50-mg; H3K4me1 Antibody (Diagenode Cat# C15410194 Lot# A1862D)
https://www.diagenode.com/en/p/h3k36me3-polyclonal-antibody-premium-50-mg; H3K36me3 Antibody (Diagenode Cat# C15410192 Lot# A1845P)
https://www.diagenode.com/en/p/h3k4me3-polyclonal-antibody-premium-50-ug-50-ul; H3K4me3 Antibody (Diagenode Cat# C15410003-50 Lot# A8034D)
https://www.diagenode.com/en/p/ctcf-polyclonal-antibody-classic-50-mg; CTCF Antibody (Diagenode Cat# C15410210-50 Lot# A2354-00234P)

# Eukaryotic cell lines

Policy information about cell lines and Sex and Gender in Research

| | |
|---|---|
| Cell line source(s) | DSMZ: https://celldive.dsmz.de/<br>ECACC: https://www.culturecollections.org.uk/<br>EverCyte:https://evercyte.com/<br>ATCC: https://www.atcc.org/<br>SW403 (ACC294, DSMZ)<br>SW480 (ACC313, DSMZ)<br>SW948 (91030714, ECACC)<br>HT29 (ACC299, DSMZ)<br>CL11 (ACC467, DSMZ)<br>C32 (12022908, ECACC)<br>HCEC-1CT (CkHT039-0229, Evercyte)<br>HEK293T (CRL-11268, ATCC) |
| Authentication | All cell lines used are well characterised and established, and recently obtained from reputable vendors. We used whole genome sequencing using NGS to perform STR profiling to authenticate our cell lines. |
| Mycoplasma contamination | Routinely checked for Mycoplasma contamination kit (LOOKOUT MYCOPLASMA PCR DETECTION KIT , Sigma Aldrich , USA) |
| Commonly misidentified lines<br>(See ICLAC register) | No commonly misidentified cell lines were used in the study |

# Plants

| | |
|---|---|
| Seed stocks | *Report on the source of all seed stocks or other plant material used. If applicable, state the seed stock centre and catalogue number. If plant specimens were collected from the field, describe the collection location, date and sampling procedures.* |
| Novel plant genotypes | *Describe the methods by which all novel plant genotypes were produced. This includes those generated by transgenic approaches, gene editing, chemical/radiation-based mutagenesis and hybridization. For transgenic lines, describe the transformation method, the number of independent lines analyzed and the generation upon which experiments were performed. For gene-edited lines, describe the editor used, the endogenous sequence targeted for editing, the targeting guide RNA sequence (if applicable) and how the editor was applied.* |
| Authentication | *Describe any authentication procedures for each seed stock used or novel genotype generated. Describe any experiments used to assess the effect of a mutation and, where applicable, how potential secondary effects (e.g. second site T-DNA insertions, mosiacism, off-target gene editing) were examined.* |

# ChIP-seq

## Data deposition

☒ Confirm that both raw and final processed data have been deposited in a public database such as GEO.

☒ Confirm that you have deposited or provided access to graph files (e.g. BED files) for the called peaks.

| | |
|---|---|
| Data access links<br>*May remain private before publication.* | https://ega-archive.org/datasets/EGAD50000000295 |
| Files in database submission | All fastq and bed files for C32, CL11, HT29, SW403, SW480, SW948 on H3K4me1,H3K4me3, H3K27ac, H3K27me3, H3K36me3 and CTCF, as well as input. Each dataset has two replicates. |
| Genome browser session<br>(e.g. UCSC) | https://genome.ucsc.edu/s/philip.law%40icr.ac.uk/CRC%20GWAS%20annotation |

## Methodology

| | |
|---|---|
| Replicates | For each cell line each anitbody capture Chipmentation experiment was performed in two replicates. This comprise six antibodies and an IgG and a Input control |
| Sequencing depth | Illumina Novaseq 6000, Single End sequencing, 100bp reads, Dual barcode (8bp,8bp), Sequencing depth varied from 30 million to 100 million reads. |
| Antibodies | All antibodies obtained from Diagenode<br># Catalog Antibody       lot<br>1.C15410196 H3K27ac Antibody - ChIP-seq Grade      A1723-0041D<br>2.C15410195 H3K27me3 Antibody - ChIP-seq Grade     A0824D<br>3.C15410194 H3K4me1 Antibody - ChIP-seq Grade      A1862D<br>4.C15410192 H3K36me3 Antibody - ChIP-seq Grade     A1845P<br>5.C15410003-50 H3K4me3 Antibody - ChIP-seq Grade   A8034D |

6. C15410210-50 CTCF Antibody - ChIP-seq Grade    A2354-0010

Peak calling parameters    MACS broad peak

Data quality    As part of the nf-core chipseq pipeline, extensive QC is performed, including adapter trimming, filtering duplicate reads and poorly mapped reads

Software    nf-core chipseq pipeline summary:
Raw read QC (FastQC)
Adapter trimming (Trim Galore!)
Alignment (BWA)
Mark duplicates (picard)
Merge alignments from multiple libraries of the same sample (picard)
  Re-mark duplicates (picard)
  Filtering to remove:
    -reads mapping to blacklisted regions (SAMtools, BEDTools)
    -reads that are marked as duplicates (SAMtools)
    -reads that arent marked as primary alignments (SAMtools)
    -reads that are unmapped (SAMtools)
    -reads that map to multiple locations (SAMtools)
    -reads containing > 4 mismatches (BAMTools)
    -reads that have an insert size > 2kb (BAMTools; paired-end only)
    -reads that map to different chromosomes (Pysam; paired-end only)
    -reads that arent in FR orientation (Pysam; paired-end only)
    -reads where only one read of the pair fails the above criteria (Pysam; paired-end only)
Alignment-level QC and estimation of library complexity (picard, Preseq)
Create normalised bigWig files scaled to 1 million mapped reads (BEDTools, bedGraphToBigWig)
Generate gene-body meta-profile from bigWig files (deepTools)
Calculate genome-wide IP enrichment relative to control (deepTools)
Calculate strand cross-correlation peak and ChIP-seq quality measures including NSC and RSC (phantompeakqualtools)
Call broad/narrow peaks (MACS2)
Annotate peaks relative to gene features (HOMER)
Create consensus peakset across all samples and create tabular file to aid in the filtering of the data (BEDTools)
Count reads in consensus peaks (featureCounts)
Differential binding analysis, PCA and clustering (R, DESeq2)
Create IGV session file containing bigWig tracks, peaks and differential sites for data visualisation (IGV).
Present QC for raw read, alignment, peak-calling and differential binding results (MultiQC, R)

