## [Peer review file · Nature Genetics]

Peer Review Information

Manuscript Title: Systematic prioritisation of functional variants and effector genes for colorectal cancer risk

Corresponding author name(s): Professor Richard (S) Houlston

Reviewer Comments & Decisions:

Decision Letter, initial version:

13th Mar 2024

Dear Professor Houlston,

First, please accept my sincere apologies for the delay in returning this decision to you - I am honestly mortified at how long it has taken. Thank you for bearing with me!

Your Article, "Systematic prioritisation of functional variants and susceptibility genes for colorectal cancer risk loci" has now been seen by 2 referees. You will see from their comments below that while they find your work of interest, some important points are raised. We are interested in the possibility of publishing your study in Nature Genetics, but would like to consider your response to these concerns in the form of a revised manuscript before we make a final decision on publication.

We therefore invite you to revise your manuscript taking into account all reviewer comments. Please highlight all changes in the manuscript text file. At this stage we will need you to upload a copy of the manuscript in MS Word .docx or similar editable format.

*2) If you have not done so already please begin to revise your manuscript so that it conforms to our Article format instructions, available

here.

*3) Include a revised version of any required Reporting Summary:

Please be aware of our guidelines on digital image standards.

[redacted]

We hope to receive your revised manuscript within four to eight weeks. If you cannot send it within this time, please let us know.

Sincerely,

Safia Danovi, PhD
Senior Editor, Nature Genetics
ORCID: 0009-0007-7822-5479

Referee expertise:

Referee #1: GWAS follow up incl. MPRA, fine mapping

Referee #2: colorectal disease genetics

Reviewers' Comments:

Reviewer #1:

Remarks to the Author:

This manuscript details a comprehensive effort to identify potentially causal variants and genes underlying colorectal cancer risk, following on from a large (>100K cases) GWAS meta-analysis of colorectal cancer (CRC) published previously in this journal. The authors first focus on identifying potentially causal risk variants through an integrative analysis combining statistical fine-mapping, new epigenomic data from CRC cell lines, functional fine-mapping (MPRA), and chromatin conformation analysis (micro-C). Specifically, they:

1. Assessed cell-type specificity of risk variants using a gene-set based approach (scDRS) and data culled from both Tabula Sapiens as well as the gut cell atlas. Reassuringly, they observed considerable enrichment for large intestine (and curiously skin), and within the gut dataset BEST4+ and colonic epithelial cells.
2. Performed comprehensive fine mapping (SUSIE&PolyFun) using functional annotations derived from ChIP (5 histone marks & CTCF) and ATACseq data generated from a panel of six CRC cell lines.
3. Functional fine-mapping of risk-associated variants using lentiviral MPRA in both two CRC cell lines as well as an immortalized primary colonic cell line.
4. Assessed enrichment of specific transcription factors in open chromatin regions (defined by ATAC data); assessed potential effects of variants on chromatin structure (Akita).
5. Integrated these data creating a tiered scoring system. Reassuringly for a large percentage of loci the lead variant was in the highest tier; alternatively, for ~25% of loci there was no variant in the highest tier.

They then focused on identification of potential causal genes using multiple methods:

1. Investigated eQTL data from normal colon (SOCCS, GTEx) and CRC (TCGA, READ) and nominating genes where MPRA-significant variants showed the same direction of effect as a QTL – nominating candidate genes for >25% of loci.
2. Activity by contact (microC/H3K27AcChIP/RNA-seq)

Finally, they explore whether these nominated genes might serve as therapeutic targets by investing multiple sources of functional and drug curations.

This work has a number of strengths:

- This follows on from a fairly large GWAS meta-analysis from one of the most common types of cancer, investigating >200 risk loci
- It is a fairly comprehensive and well-done multi-omics based effort to identify candidate causal variants and genes for CRC; most of these loci have not been investigated in depth and this work provides many leads for in-depth work exploring mechanisms underlying risk, tumor development, and or therapeutic targeting of tumors
- A large body of mostly genome-scale data that could be utilized by others studying CRC (Histone/CTCF ChIP & ATACseq x 6 CRC cell lines, microC x 5 CRC cell lines, MPRA x 8,000 variants in 3 cell lines)
- Nominated potential causal genes for a large proportion of risk loci (at least one gene is nominated

for 170 loci)

- The approach used seems to have a nice balance between sensitivity and specificity for identifying potential target genes.
- The manuscript is well-written and clear.

I find it a little surprising that of the 208 genes nominated so few (12!) have not previously been proposed to influence CRC risk. Where do these annotations come from that lead to this conclusion? Could this be attributable to that quite frequently the authors analysis is nominating the nearest gene as the best gene, so the GWAS itself was the main source proposing those genes as plausible risk genes? Or is so widely studied that most genes are proposed to influence CRC?

One issue that is frequently raised in review of large GWAS themselves is that despite all this work and pathway analyses, this study really hasn't itself firmly uncovered novel mechanisms underlying risk or tumor development. That said – this is a very compressive investigation and is a major next step providing leads for the study of many loci. While this type of manuscript doesn't appear in this journal very frequently, this type of well-done and comprehensive work is absolutely crucial for translating GWAS findings into new biological understanding.

I don't find major issues with this paper; there are a few minor issues that the authors might address.

1. Data availability: the data generated for this paper are a valuable resource to the research community. The authors note in the data availability section that they are depositing cell line data in EGA. Can they be specific about which data (all of it?) and in what form(s): e.g. sequence reads, pre-analyzed data, etc.?

2. The authors perform fine-mapping using SUSIE and PolyFun, weighting probabilities using a set of functional annotations provided by the Alkes Price group supplemented with in-house (new) histone ChIP and ATAC seq data from six CRC cell lines. Having already assessed cell-type enrichment of histone marks using RoadMap data derived from non-malignant cells, is there a reason these histone marks were not also included in fine-mapping? To very clear: I am not asking that fine-mapping be re-done with additional annotation data from Roadmap – I feel it's beyond unreasonable to ask as the results may subtly change and change things downstream, and frankly I don't think this is likely to dramatically change the authors conclusions. Is there a way the authors could use data from CRC cells (or tumors if available) to compare cell-specificity between normal and tumor so a reader can assess the relative strengths and weaknesses of fine-mapping with CRC-focused annotations?

3. Page 5, 2nd paragraph, "MPRA-significant variants preferentially localized to open chromatin ($p=...$) and mapped to the transcription start sites of genes associated with chromatin interactions ($p=...$).": It isn't clear to me what the authors mean by the last half of this sentence. I think the authors are suggesting that MPRA significant variants tend to more often interact with the TSS of genes, but I read this as they are located in/near gene TSSs. Can this be clarified?

4. Table ST5 (SMR): I realize this is output from SMR, but it would be helpful to define some of the column headings, a few of which aren't particularly intuitive. This is perhaps an issue for a couple of the other supplementary tables?

5. The micro-C analysis and annotation in ST6 aren't entirely clear. The legend in ST6 notes micro-C annotations as "Micro-C TSS: GWAS locus region has a Micro-C contact with the TSS of a gene; Micro-

C: GWAS locus region has a Micro-C contact with the body of a gene". This implies that there is a connection with the TSS or body of a gene from anywhere in the region of association. Are the authors actually annotating interactions only where the Tier 1 variant itself specifically falls within the region of significant interaction (which is what seems to be implied in the manuscript; the examples in the Supplementary Figures seem to be consistent with this)? This should be clarified and explicit.

6. 30% of the nominated variants were outside the TAD of the target gene – perhaps the authors could comment on why this might be?

7. Supplementary Figure 3: the enrichment the authors point out in the manuscript is clear, however the p-value coloring scale seems to not really distinguish differences at the upper levels of $-\log_{10}P$ for a few histone marks, particularly for H3K4Me1. It might be useful to alter the coloring or perhaps list the results as a supplementary table?

8. The MPRA design seems sensible, indeed it's a strength that it didn't rely entirely on SUSIE/POLYFUN and was instead much more conservative/inclusive. It isn't entirely clear in the MPRA design whether the design made effort to assess variants in LD to lead signals that were not in the summary statistics (e.g. that were either not imputed in the CRC meta-analysis because they weren't in the imputation reference or alternatively failed imputation QC and thus not fine-mappable). Obviously the MPRA has already been run and this would be a relatively small number of variants, but it would be nice to note this in the methods or as a potential limitation?

Reviewer #3:

Remarks to the Author:

The authors have generated a large and highly relevant set of datasets to help understand the function of colon cancer risk variants. These datasets include multiplex reporter assays on over 8,000 potential causal variants to determine enhancer activity in cancer and "normal" tissue cell lines, along with assays of chromatin contacts with Micro C, open chromatin with Omni-ATAC and epigenetic marks with ChIPmentation.

These are combined with analyses of external data, including fine-mapping of GWAS summary statistics and integration with eQTL data, to attempt to identify and understand risk variants and genes for colon cancer.

The generation of datasets is very well-chosen, and I felt it was quite a useful display of how much high-information functional data can be generated by a single group using the newest generation of functional genomics experiments.

The authors essentially summarize their work into two measures - a relatively complex scoring system for ranking potential colon cancer risk variants (ST4), and a somewhat simpler rule-of-thumb approach for linking these risk variants to genes (ST6). They use these to highlight various potentially novel findings, including in possible drug targets.

Neither the experimental approaches nor the analysis approaches are entirely novel (see my major comment below with references about more fully citing previous work). However, to my knowledge the application to colon cancer is novel. In addition, generating a large and comprehensive dataset

across different data modalities targeted at the genetics of one disease potentially has great value. My other major comments are about more clearly demonstrating the value of these datasets, and in ensuring the methods are clearly described.

Major comment 1: Introduction and prior work

The introduction is very brief, and would benefit from taking more time to discuss prior work on using these sorts of functional experiments for interpreting risk variants. The authors already cite Long et al's (PMID:36423637) work on using MPRA for melanoma risk variants, which was clearly an inspiration, but not until half way through the results. Discussing this paper, and Choi et al (PMID:32483191) from the same group, in the discussion would help set the scene, as would citing other MPRA analyses across other traits like PMID:35298243. Likewise, discussing previous work using Micro-C/Micro Capture-C etc for studying risk variants (e.g. work in covid PMID:34737427, review PMID:36340032), and similar work on ATAC-Seq/Chipmentation (e.g. PMID:31548716) would also help put the work in context.

Major comment 2: Demonstrating utility of variant and gene prioritization

I think that the datasets the authors have generated are clearly of value to the field in general. However, it is not clear to me that the authors have cleanly demonstrated that their approach, end-to-end, has added significant value in prioritizing genes and variants compared to simpler approaches.

I would like to see some more systematic analysis of the variant prioritization score used, compared to e.g. a normal fine-mapping analysis. Does this method produce a demonstrably higher quality set of variants? Are they more strongly enriched for known risk variants? Do they overlap more closely with eQTL variants, or with fine-mapped variants for other cancers?

I was not terribly convinced by the authors citing the scoring metrics success on two known associations in the discussion. The MYC causal variant (rs6983267) is the highest ranked in the region in ST4 (and is supported by MPRA evidence), but it is also equally well supported by a simple fine-mapping. By contrast the causal variant at POU2AF2 cited in Rajasekaran et al (rs3087967) is only the joint-10th best rated variant in that locus according to ST4 with a score of 9 (and with no supporting evidence from MPRA), much lower than the highest-rated variant in the region (rs7130173, with a score of 15). This does not seem like clear evidence that the additional information is adding significant value to picking causal variants.

Likewise, for the candidate gene sets produced, I would like to be convinced that the genes selected are clearly higher quality than those produced by simpler approaches, e.g. than the set of genes that would be selected using just publicly available data like the Open Targets L2D score uses, or even than just choosing the nearest gene. Again, the authors could assess these using similar approaches to evaluation as the variants (are they more strongly enriched for known risk pathways, or known oncogenes, etc).

Major comment 3: Clarity of methods descriptions

There were a number of places in the methods where I was not able to understand what had been done.

For example, I found it very hard to follow the description of the statistical fine-mapping in the methods. This is a relatively standard analysis, and so I can follow the gist, but I was confused on some points, and I imagine a non-specialist would find it very hard to follow. The authors should make sure that the input and output of each stage is clearly described (e.g. what went into PolyFun, what did it produce, what was passed on to SusieR, how was the output processed, etc).

To pull out some specific issues:

The authors state: "We incorporated functional annotation by calculating the heritability of each variant, weighted by functional annotation." I am not clear what this refers to. Does this refer to per-SNP priors produced by PolyFun (SNPVAR)? Or is it something the authors calculated themselves?

The authors write: "Variants with PIP > 0.1 were considered eligible for being included in a credible set." I do not know what this means in practice, and it seems to contradict the results which say that there were credible sets with 226 variants (which couldn't all have PIP > 0.1, as this would sum to more than 1).

I am also not clear on how the authors handled independent signals during the fine-mapping - independent signals are mentioned in many places, and it seems like a conditional analysis was done prior to fine-mapping (line 467), but exactly how this was done and then how the output was combined into credible sets across independent signals is not clear.

Presumably for SusieR and any conditional analysis an LD matrix will be required. What LD matrix was used, and how was it matched to the ancestry of the GWAS set?

Similarly, I found the description of how variants were prioritized for MPRA hard to follow as well. The text states:

"We selected variants mapping to the CRC risk loci for MPRA testing according to the following process: (1) variants with a log likelihood ratio (LLR) < 10^{-3} and <250kb relative to the primary lead SNP and with a P-value < 10^{-5} ; (2) variants within 250kb of a lead variant and with LD r^2 266 < 0.2 were considered secondary loci and treated as per step (1)."

I have reread this a number of times, going back and forth with table ST4, and I am unable to understand what was done. What is the log likelihood ratio (the ratio between what and what)? There is a LLR column in ST4, but it is almost never 10^{-3}. If you keep repeating this process until you have included all variants with $p < 1e-5$, how is this different from just including all variants with $p < 1e-5$? Is this a process to pick variants, or to assign variants to LD clusters? And, perhaps most importantly, if the authors have already carried out statistical fine-mapping to establish likely causal variants and independent signals, why are they using this entirely different rule to pick variants to assay rather than just assaying variants in the 95% credible sets (or 99% credible sets, if they wanted to do more variants)?

Minor comments

- I was not able to find the results of the fine-mapping in the supplementary tables - I can figure some of it out from ST4, but the authors should really put the full results, with PIPs etc, in the supplementary data.

- At line 102, the authors describe which TFs most commonly bind the risk variants. This is confounded by how widely this transcription factor binds in the first place (CFCT, for instance, is one of the most active TFs in most tissues). It would be informative to include an enrichment adjusted for total target size (e.g. proportion of loci bound by TF compared to proportion of control regions bound).

- On page 6, the authors state "79 (29%) of the MPRA-significant variants at 27 loci displayed a consistent direction of effect between MPRA and eQTL". I am not sure I follow exactly what this means. It might be better to describe how many loci had both MPRA and eQTL effects, and of those how many had consistent vs opposite directions of effect. It would also be helpful to capture this information, e.g. as an extra column in ST5.

Author Rebuttal to Initial comments

Reviewer #1:

I find it a little surprising that of the 208 genes nominated so few (12!) have not previously been proposed to influence CRC risk. Where do these annotations come from that lead to this conclusion? Could this be attributable to that quite frequently the authors analysis is nominating the nearest gene as the best gene, so the GWAS itself was the main source proposing those genes as plausible risk genes? Or is so widely studied that most genes are proposed to influence CRC?

Our statement in the original text regarding the number of genes purported to underlie GWAS signals was simply based on collating what had been stated in GWAS publications, whether or not there was any supporting evidence to implicate the gene (often it was simply the closest gene). We fully acknowledge that in retrospect this was not informative, and indeed, also paradoxically undermines our work. Responding to the reviewer, we now report a formal investigation of which of the candidate target genes we implicate in CRC predisposition have supporting evidence for their role in CRC biology. Specifically, we conducted text mining of the available biomedical literature using the OncoScore tool (Rocco *et al*, 2017 Sci Rep. PMID: 28387367). We complemented this analysis by also querying semantic predications within the Semantic MEDLINE Database (SemMedDB), which is based on all PubMed citations, using MELODI Presto (Elsworth and Gaunt, 2021, Bioinformatics, PMID: 32810207). Based on the intersection of the output from both OncoScore and SemMedDB we now state: "To determine which of the target genes we identified are already known to have a role in CRC, and more broadly cancer, we used the text mining tool OncoScore, which is based on all available biomedical literature.

To complement this analysis we queried semantic predications within the Semantic MEDLINE Database using MELODI Presto. Intersecting the results from these searches, 142 of the 208 candidate target genes we identify appear to have no documented role in CRC and 33 of these presently have no established role in any cancer (Supplementary Table 7 and 8)."

One issue that is frequently raised in review of large GWAS themselves is that despite all this work and pathway analyses, this study really hasn't itself firmly uncovered novel mechanisms underlying risk or tumor development. That said – this is a very compressive investigation and is a major next step providing leads for the study of many loci. While this type of manuscript doesn't appear in this journal very frequently, this type of well-done and comprehensive work is absolutely crucial for translating GWAS findings into new biological understanding.

The primary motivation for us undertaking the study was to undertake a comprehensive investigation of each of the GWAS risk loci for colorectal cancer, and we appreciate that the reviewer recognises our efforts. While we fully understand that in any study it's always gratifying to uncover a novel mechanism, the very nature of science dictates that this is not a given. In revising our designation of what constitutes a novel candidate target gene based on formal text mining of the literature, we now provide evidence to implicate a number of genes as novel targets.

I don't find major issues with this paper; there are a few minor issues that the authors might address.

1. Data availability: the data generated for this paper are a valuable resource to the research community. The authors note in the data availability section that they are depositing cell line data in EGA. Can they be specific about which data (all of it?) and in what form(s): e.g. sequence reads, pre-analyzed data, etc.?

We fully appreciate the value of making data publicly accessible and all of the data generated on all the cell lines has been submitted to EGA under the following accessions: EGAD5000000294 (Micro-C), EGAD5000000295 (ChIP-seq, all marks), EGAD5000000296 (ATAC-seq), EGAD5000000297 (RNA-

seq). We provide both the raw FASTQ files and final processed data (e.g. bed files for CHIP-seq, cooler files for Micro-C).

2. The authors perform fine-mapping using SUSIE and PolyFun, weighting probabilities using a set of functional annotations provided by the Alkes Price group supplemented with in-house (new) histone CHIP and ATAC seq data from six CRC cell lines. Having already assessed cell-type enrichment of histone marks using RoadMap data derived from non-malignant cells, is there a reason these histone marks were not also included in fine-mapping? To very clear: I am not asking that fine-mapping be re-done with additional annotation data from Roadmap – I feel it's beyond unreasonable to ask as the results may subtly change and change things downstream, and frankly I don't think this is likely to dramatically change the authors conclusions. Is there a way the authors could use data from CRC cells (or tumors if available) to compare cell-specificity between normal and tumor so a reader can assess the relative strengths and weaknesses of fine-mapping with CRC-focused annotations?

We analysed the Roadmap data across cell types solely to demonstrate cell-specificity. Having shown enrichment of associations with colorectal tissue we thereafter performed the fine-mapping using our in-house generated data on colorectal cancer cell lines. We acknowledge that the addition of the Roadmap colon data is unlikely to materially change the results from the fine-mapping.

3. Page 5, 2nd paragraph, "MPRA-significant variants preferentially localized to open chromatin ($p=...$) and mapped to the transcription start sites of genes associated with chromatin interactions($p=...$).": It isn't clear to me what the authors mean by the last half of this sentence. I think the authors are suggesting that MPRA significant variants tend to more often interact with the TSS of genes, but I read this as they are located in/near gene TSSs. Can this be clarified?

We apologise for any ambiguity. We have now revised our text to explicitly state: "MPRA-significant variants preferentially localised to open chromatin ($P = 7.32 \times 10^{-35}$) as well as mapped to regions that interacted with the TSS of genes through a Micro-C chromatin interaction".

4. Table ST5 (SMR): I realize this is output from SMR, but it would be helpful to define some of the column headings, a few of which aren't particularly intuitive. This is perhaps an issue for a couple of the other supplementary tables?

As requested we now provide additional information to the legends for Supplementary Tables 4 (reordered to Supplementary Table 2), 5, and 6.

5. The micro-C analysis and annotation in ST6 aren't entirely clear. The legend in ST6 notes micro-C annotations as "Micro-C TSS: GWAS locus region has a Micro-C contact with the TSS of a gene; Micro-C: GWAS locus region has a Micro-C contact with the body of a gene". This implies that there is a connection with the TSS or body of a gene from anywhere in the region of association. Are the authors actually annotating interactions only where the Tier 1 variant itself specifically falls within the region of significant interaction (which is what seems to be implied in the manuscript; the examples in the Supplementary Figures seem to be consistent with this)? This should be clarified and explicit.

The reviewer is correct in the interpretation. For the variant to gene association, we wanted to only use the data for the Tier 1 variants, *i.e.* the best annotated variants. Thus, for both the "Micro-C TSS" and "Micro-C" annotations, we checked if Tier 1 variants fall in a region that interacts with a gene TSS or gene body through a Micro-C interaction, respectively. To avoid any ambiguity we have revised the legend of Supplementary Table 6 to include this information, as well as revising the accompanying text.

6. 30% of the nominated variants were outside the TAD of the target gene – perhaps the authors could comment on why this might be?

While the majority of interactions take place within TADs it has been long been recognised that significant and strong long-range interactions occur outside TADs (*e.g.* Smith *et al.* 2016, AJHG, PMID:

26748519). It is conceivable that indirect effects may occur, *i.e.* the SNP might affect the expression of a regulatory gene within its TAD, which in turn impacts the target gene in another TAD.

7. Supplementary Figure 3: the enrichment the authors point out in the manuscript is clear, however the p-value coloring scale seems to not really distinguish differences at the upper levels of $-\log_{10}P$ for a few histone marks, particularly for H3K4Me1. It might be useful to alter the coloring or perhaps list the results as a supplementary table?

All the elements in the heatmap in dark blue have P -values equal to 2×10^{-5} , based on 50,000 permutations (*i.e.* corresponding to the smallest P -value).

8. The MPRA design seems sensible, indeed it's a strength that it didn't rely entirely on SUSIE/POLYFUN and was instead much more conservative/inclusive. It isn't entirely clear in the MPRA design whether the design made effort to assess variants in LD to lead signals that were not in the summary statistics (e.g. that were either not imputed in the CRC meta-analysis because they weren't in the imputation reference or alternatively failed imputation QC and thus not fine-mappable). Obviously the MPRA has already been run and this would be a relatively small number of variants, but it would be nice to note this in the methods or as a potential limitation?

We did not investigate variants that were not in the summary statistics or reference panels. As mentioned, a disadvantage of the MPRA design requires that the variants be predefined ahead of the experiment. In this instance, we used the P -value from the meta-analysis, in conjunction with LD to the lead SNPs to prioritise the variants to investigate. We have expanded the methods describing the variant selection for the MPRA. We acknowledge the reviewer's comments regarding potential limitations of having to pre-select variants in our revised text.

Reviewer #3:

Remarks to the Author:

Major comment 1: Introduction and prior work

The introduction is very brief, and would benefit from taking more time to discuss prior work on using these sorts of functional experiments for interpreting risk variants. The authors already cite Long et al's (PMID:36423637) work on using MPRA for melanoma risk variants, which was clearly an inspiration, but not until half way through the results. Discussing this paper, and Choi et al (PMID:32483191) from the same group, in the discussion would help set the scene, as would citing other MPRA analyses across other traits like PMID:35298243. Likewise, discussing previous work using Micro-C/Micro Capture-C etc for studying risk variants (e.g. work in covid PMID:34737427, review PMID:36340032), and similar work on ATAC-Seq/Chipmentation (e.g. PMID:31548716) would also help put the work in context.

We appreciate the reviewer's suggestion regarding how the introduction section should be expanded and might be structured, and have revised our text accordingly. Specifically, we now discuss prior work on the strategies for the identification of causal variants and target genes underlying GWAS signals including appropriate references.

Major comment 2: Demonstrating utility of variant and gene prioritization

I think that the datasets the authors have generated are clearly of value to the field in general. However, it is not clear to me that the authors have cleanly demonstrated that their approach, end-to-end, has added significant value in prioritizing genes and variants compared to simpler approaches.

I would like to see some more systematic analysis of the variant prioritization score used, compared to e.g. a normal fine-mapping analysis. Does this method produce a demonstrably higher quality set of variants? Are they more strongly enriched for known risk variants? Do they overlap more closely with eQTL variants, or with fine-mapped variants for other cancers?

It is difficult to separate the statistical fine-mapped variants from the variants prioritised here (*i.e.* the Tier 1 variants) as the fine-mapping results make up part of the scoring. Similarly for the eQTL and functional data, since these metrics are part of the scoring, they will by their nature be enriched for these features. Our approach is analogous to work reported by, amongst others, Long *et al* (Am J Hum Genet. 2022, PMID: 36423637) and Fachal *et al* (Nat Genet 2020, PMID: 31911677) in which functional annotation data is used to prioritise variants.

I was not terribly convinced by the authors citing the scoring metrics success on two known associations in the discussion. The MYC causal variant (rs6983267) is the highest ranked in the region in ST4 (and is supported by MPRA evidence), but it is also equally well supported by a simple fine-mapping. By contrast the causal variant at POU2AF2 cited in Rajasekaran *et al* (rs3087967) is only the joint-10th best rated variant in that locus according to ST4 with a score of 9 (and with no supporting evidence from MPRA), much lower than the highest-rated variant in the region (rs7130173, with a score of 15). This does not seem like clear evidence that the additional information is adding significant value to picking causal variants.

While we acknowledge the point about the 8q24 (MYC) variant, it is not always the case that the top fine-mapped variant is the functional variant. Additionally, for several loci, there are multiple variants for the credible set. Of the 170 loci, there are only 38 where there is a single fine-mapped variant, and in several of these there are multiple credible sets with an equally significant PIP (e.g. rs12973410 and rs384991 both are in LD with rs34797592, each in their own credible sets, and both have a PIP of 1).

While *POU2AF2* is the target gene, as the reviewer rightly points out, in Rajasekaran *et al* its expression is linked to the rs3087967 genotype. rs7130173 is in strong LD with rs3087967 ($r^2 = 0.98$, $D' = 1.00$) and both SNPs display a similarly strong *POU2AF2* eQTL. The paper by Rajasekaran *et al* did not seek to identify the causal SNP underlying the 11q23.1 association as this was not the focus of the study but simply anchored the analysis to rs3087967 as it was the SNP reported in the GWAS meta. To avoid confusion, we have revised our text by deleting reference to this.

Likewise, for the candidate gene sets produced, I would like to be convinced that the genes selected are clearly higher quality than those produced by simpler approaches, e.g. than the set of genes that would be selected using just publicly available data like the Open Targets L2D score uses, or even than just choosing the nearest gene. Again, the authors could assess these using similar approaches to evaluation as the variants (are they more strongly enriched for known risk pathways, or known oncogenes, etc).

Herein, we were not simply aiming to propose gene targets but to provide functional evidence to both identify functional variants and support target gene candidacy, rather than relying on statistical modelling as is common. We hope that our revised introduction which was suggested provides the rationale for our endeavour.

As the reviewer points out there are many ways of nominating target genes. We assume the reviewer was referring to OpenTarget's V2G (variant to gene) analysis. For the reviewer's interest, using this tool the majority of the V2G scores were relatively low (median 0.28), and if we filtered to those with a V2G score > 0.3 this tool only leads to the identification of 72 genes for the 170 risk loci. As it's increasingly apparent from ENCODE etc (e.g. Gschwind et al, 2023, bioRxiv, doi:10.1101/2023.11.09.563812) enhancer-gene relationships are tissue-specific, and was the reason we based our analysis on data generated from colonic cells. However, V2G uses information from haematopoietic cells, which may therefore account for its apparent poorer performance when applied to CRC GWAS data.

To the extent they have been studied, most cancer GWAS signals do not necessarily implicate a driver gene for the specific cancer as being the target gene underlying the GWAS risk locus. We compared the gene lists to the driver genes catalogued by IntoGen (<https://www.intogen.org/>); 3 driver genes (*SMAD3*, *TCF7L2*, *CTNNB1*) in the closest gene list, 1 driver gene (*HLA-A*) in the OpenTargets V2G gene list, and 6 genes (*SMAD3*, *TCF7L2*, *BCL9L*, *CDH1*, *SOX9*, *TBX3*) in our integrative scoring gene list.

For completeness we also investigated OpenTarget's L2G (locus to gene) tool, but the results implementing the latest CRC GWAS meta-analysis are not yet available. Using the data from the

previous GWAS meta-analysis of CRC that is available in OpenTargets (Law et al, 2019, NComms, PMID: 31089142), there is a confident L2G prediction for 52 genes.

Major comment 3: Clarity of methods descriptions

There were a number of places in the methods where I was not able to understand what had been done. For example, I found it very hard to follow the description of the statistical fine-mapping in the methods. This is a relatively standard analysis, and so I can follow the gist, but I was confused on some points, and I imagine a non-specialist would find it very hard to follow. The authors should make sure that the input and output of each stage is clearly described (e.g. what went into PolyFun, what did it produce, what was passed on to SusieR, how was the output processed, etc).

We have added additional information to the methods, to further describe the processes taken to perform the analyses.

To pull out some specific issues:

The authors state: "We incorporated functional annotation by calculating the heritability of each variant, weighted by functional annotation." I am not clear what this refers to. Does this refer to per-SNP priors produced by PolyFun (SNPVAR)? Or is it something the authors calculated themselves?

We performed PolyFun analysis as advocated by the authors, specifically with reference to the section "PolyFun approach 3: Computing prior causal probabilities non-parametrically" (<https://github.com/omerwe/polyfun/wiki/1.-Computing-prior-causal-probabilities-with-PolyFun#polyfun-approach-3-computing-prior-causal-probabilities-non-parametrically>). We have revised our text to clarify this.

The authors write: "Variants with PIP > 0.1 were considered eligible for being included in a credible set." I do not know what this means in practice, and it seems to contradict the results which say that there were credible sets with 226 variants (which couldn't all have PIP > 0.1, as this would sum to more than 1).

We thank the reviewer for bringing this error in the text to our attention. The correct value is 0.001 (0.1%), and is the default value implemented in PolyFun.

I am also not clear on how the authors handled independent signals during the fine-mapping - independent signals are mentioned in many places, and it seems like a conditional analysis was done prior to fine-mapping (line 467), but exactly how this was done and then how the output was combined into credible sets across independent signals is not clear.

The conditional analysis was performed as part of our original meta-analysis (Fernandez-Rozadilla 2023, Nat. Genet, PMID: 36539618). We have included the methodology in the manuscript. Independent variants identified by the conditional analysis were considered separately throughout the analysis.

Presumably for SusieR and any conditional analysis an LD matrix will be required. What LD matrix was used, and how was it matched to the ancestry of the GWAS set?

As the GWAS data were based on east Asian and European individuals, as a reference for LD estimation, we made use of genotyping data from 6,684 unrelated samples of east Asian ancestry and 4,284 samples from combined UK10K and European samples in 1000 Genomes. Ancestry was determined using the provided data from the respective projects. The conditional analysis was performed on each ancestry separately, and combined using a meta-analysis.

The fine-mapping was performed using non-cancer individuals from the Genomics England dataset (45,498 European individuals, <https://re-docs.genomicsengland.co.uk/aggv2/>), as a dataset in GRCh38 was required. These individuals' ancestry was ascertained by a PCA based on data from the 1,000 Genomes Project phase 3.

Similarly, I found the description of how variants were prioritized for MPRA hard to follow as well. The text states:

"We selected variants mapping to the CRC risk loci for MPRA testing according to the following process: (1) variants with a log likelihood ratio (LLR) $< 10^{-3}$ and $< 250\text{kb}$ relative to the primary lead SNP and with a P-value $< 10^{-5}$; (2) variants within 250kb of a lead variant and with LD $r^2 > 0.2$ were considered secondary loci and treated as per step (1)."

I have reread this a number of times, going back and forth with table ST4, and I am unable to understand what was done. What is the log likelihood ratio (the ratio between what and what)? There is a LLR column in ST4, but it is almost never $< 10^{-3}$. If you keep repeating this process until you have included all variants with $p < 1e-5$, how is this different from just including all variants with $p < 1e-5$? Is this a process to pick variants, or to assign variants to LD clusters? And, perhaps most importantly, if the authors have already carried out statistical fine-mapping to establish likely causal variants and independent signals, why are they using this entirely different rule to pick variants to assay rather than just assaying variants in the 95% credible sets (or 99% credible sets, if they wanted to do more variants)?

We thank the reviewer for identifying this error. The correct statement is “log₁₀ likelihood ratio (LLR) < 3 ” (equivalent to a likelihood ratio $< 10^{-3}$). The $P < 10^{-5}$ was a filter, where any variants with a P-value greater than this were not considered. We also identified an omission in the methodology and have expanded the methods to clarify. We now state: “We selected variants mapping to the CRC risk loci for MPRA testing by first considering all SNPs within a 500kb window of the primary or conditional association, imposing a log likelihood ratio ($\log_{10}(P_{\text{variant}}/P_{\text{lead SNP}})$) < 3.0 . As this might not capture functional variants which are remain highly significant at some loci (*i.e.* where the lead SNP has an extremely strong association) we also considered SNPs having $P_{\text{GWAS}} < 0.7 * -\log_{10}(P_{\text{lead snp}})$ as well as stipulating an $r^2 > 0.2$ and $P_{\text{GWAS}} < 10^{-5}$ ”. The value of 0.7 was constrained by the capacity of the oligonucleotide synthesis chip.

Minor comments

- I was not able to find the results of the fine-mapping in the supplementary tables - I can figure some of it out from ST4, but the authors should really put the full results, with PIPs etc, in the supplementary data.

We now include the fine-mapping results (PIP and credible set) in Supplementary Table 2 (previously Supplementary Table 4).

- At line 102, the authors describe which TFs most commonly bind the risk variants. This is confounded by how widely this transcription factor binds in the first place (CFCT, for instance, is one of the most active TFs in most tissues). It would be informative to include an enrichment adjusted for total target size (e.g. proportion of loci bound by TF compared to proportion of control regions bound).

We acknowledge the point and have now included an enrichment analysis comparing the GWAS regions, to regions with no GWAS signal (Supplementary Fig. 5). We performed a similar analysis to the Roadmap tissue specificity analysis, where the tally of TFs bound in the GWAS regions was compared to randomly selected regions with no GWAS signal ($P_{GWAS} > 0.95$) of equivalent size. The process was repeated 50,000 times. The TFs stated in the text (CTCF, ZNF460, PRDM9, SP1, KLF5), as well as several others were enriched ($P < 10^{-3}$).

- On page 6, the authors state "79 (29%) of the MPRA-significant variants at 27 loci displayed a consistent direction of effect between MPRA and eQTL". I am not sure I follow exactly what this means. It might be better to describe how many loci had both MPRA and eQTL effects, and of those how many had consistent vs opposite directions of effect. It would also be helpful to capture this information, e.g. as an extra column in ST5.

As suggested we now state how many loci had both MPRA and eQTL effects, and of those how many had consistent directions of effect. Specifically: "Of the 275 MPRA-significant variants, 113 had a significant eQTL ($P_{eQTL} < 0.05/665$; correcting for the number of unique genes tested). Of these, 79

displayed a consistent direction of effect between MPRAs and eQTL". We also provide this information in Supplementary Table 4

Decision Letter, first revision:

10th Jun 2024

Dear Professor Houlston,

First, I'm so sorry that it's taken such a long time to send you this decision. Thank you for bearing with me.

Your Article, "Systematic prioritisation of functional variants and susceptibility genes for colorectal cancer risk loci" has now been seen by 2 referees. You will see from their comments below that while they find your work of interest, some important points are raised. We are interested in the possibility of publishing your study in Nature Genetics, but would like to consider your response to these concerns in the form of a revised manuscript before we make a final decision on publication.

Our intention is to assess the revised manuscript in-house but, depending on your response, we might have to return to one or both reviewers. Please be assured that we are keen to avoid incurring further reviews and will only do this if we deem it necessary.

We therefore invite you to revise your manuscript taking into account all reviewer and editor comments. Please highlight all changes in the manuscript text file. At this stage we will need you to upload a copy of the manuscript in MS Word .docx or similar editable format.

*2) If you have not done so already please begin to revise your manuscript so that it conforms to our Article format instructions, available here.

*3) Include a revised version of any required Reporting Summary:

It will be available to referees (and, potentially, statisticians) to aid in their evaluation if the

manuscript goes back for peer review.
A revised checklist is essential for re-review of the paper.

Please be aware of our guidelines on digital image standards.

[redacted]

We hope to receive your revised manuscript within four to eight weeks. If you cannot send it within this time, please let us know.

Sincerely,

Safia Danovi, PhD
Senior Editor, Nature Genetics
ORCID: 0009-0007-7822-5479

Reviewers' Comments:

Reviewer #1:

Remarks to the Author:

I feel this submission adequately addressed most of the issues I raised previously.

I have one remaining issue. While I appreciate the authors making the data they generated publicly available, I note however that the MPRA data doesn't seem to be included and should be. As it stands,

only a very basic summary of results are provided, and only for variants that are significant in at least one of the three cell lines tested (ST4)

For ST4, no alleles are listed so it's impossible to tell from the table what the direction of effect is, alleles should be included in the table.

Beyond this summary which is useful to the reader, these data should be made at a finer level, preferably such that they could be reanalyzed.

Reviewer #3:

Remarks to the Author:

#Major comment 1: Introduction and prior work

The new introduction sets the scene much better.

#Major comment 2: Demonstrating utility of variant and gene prioritization

The authors are correct that many other publications deploy novel prioritization approaches to identify candidate variants or genes in GWAS loci without systematically assessing the extent to which they add predictive value or biological insight beyond simpler approaches. I think that this has become somewhat of a problem in the literature - we are starting to see a profusion of automatic and bespoke variant- and gene-lists without clear evidence of which are the most effective. This does not make them valueless, but it does place a burden on the reader to make difficult subjective judgements about which information is or is not valuable. I did not review the papers that the authors cite, and if I had I would also have asked for the authors to provide some data on incremental value of the approach for those papers.

In the case of this specific paper under review, where it leaves us with is a collection of clearly useful and valuable information on colorectal cancer risk variants, which are then synthesised into combined variant and gene lists for which I do not feel I am able to assess/quantify the incremental value (in terms of) over just e.g. picking the nearest gene or, more powerfully, Open Target's L2G genes. I do appreciate the authors' point that this list is more tractable to the reader because it includes data on specific functional links rather than just a block-black statistical posterior (though I think this argument applies primary to the underlying functional data and not to the combined scores or ranks, which are harder to interpret).

Anyway, this review is clearly not the place to draw a line in the sand on this issue. If the authors do not want to investigate the incremental value further, and the editor does not wish to push it, then I will not push any further.

#Major comment 3: Clarity of methods descriptions

The methods are improved, but I still find it very difficult to follow exactly how the variants for MPRA were picked, and I think there are still errors or confusingly written parts. I would appreciate the authors taking another pass. Specific issues to address:

- There is an inconsistency between what has been written in the point-to-point response and in the actual paper. The paper uses " $0.7 \cdot -\log_{10}(\text{P-value})$ " and the P2P uses " $\text{P_GWAS} < 0.7 \cdot -\log_{10}(\text{P_lead snp})$ ". Neither seems right. Are the authors trying to say " $-\log_{10}(\text{P_GWAS}) < 0.7 \cdot -\log_{10}(\text{P_lead snp})$ "?
- The use of the term "log likelihood ratio" to mean "log ratio of p-values" is confusing (as likelihood ratio has a specific meaning in hypothesis testing, which is not how the authors seem to be using it here), and should be changed for something more descriptive
- The text uses "p_GWAS" and "p_variant" - do these mean different things?
- Does " $r^2 > 0.2$ " mean " $r^2 > 0.2$ with at least one lead risk variant"?
- Does this r^2 threshold, and the " $\text{P_GWAS} < 10e-5$ " threshold, apply to all variants, or just the variants that meet the $0.7 \cdot \log(\text{p-value})$ criteria?

Author Rebuttal, first revision:

Reviewer #1:

Remarks to the Author:

I feel this submission adequately addressed most of the issues I raised previously.

I have one remaining issue. While I appreciate the authors making the data they generated publicly available, I note however that the MPRA data doesn't seem to be included and should be. As it stands, only a very basic summary of results are provided, and only for variants that are significant in at least one of the three cell lines tested (ST4)

For ST4, no alleles are listed so it's impossible to tell from the table what the direction of effect is, alleles should be included in the table.

Beyond this summary which is useful to the reader, these data should be made at a finer level, preferably such that they could be reanalyzed.

Response: We apologise for the omission. All of the MPRA data has now been uploaded to EGA under accession EGAD50000000596, as both FASTQ files and the final output from MPRAalyze. We have included the allele information in ST4.

Reviewer #3:

Remarks to the Author:

#Major comment 1: Introduction and prior work

The new introduction sets the scene much better.

Response: We thank the reviewer for acknowledging our efforts.

#Major comment 2: Demonstrating utility of variant and gene prioritization

The authors are correct that many other publications deploy novel prioritization approaches to identify candidate variants or genes in GWAS loci without systematically assessing the extent to which they add predictive value or biological insight beyond simpler approaches. I think that this has become somewhat of a problem in the literature - we are starting to see a profusion of automatic and bespoke variant- and gene-lists without clear evidence of which are the most effective. This does not make them valueless, but it does place a burden on the reader to make difficult subjective judgements about which information is or is not valuable. I did not review the papers that the authors cite, and if I had I would also have asked for the authors to provide some data on incremental value of the approach for those papers.

In the case of this specific paper under review, where it leaves us with is a collection of clearly useful and valuable information on colorectal cancer risk variants, which are then synthesised into combined variant and gene lists for which I do not feel I am able to assess/quantify the incremental value (in terms of) over just e.g. picking the nearest gene or, more powerfully, Open Target's L2G genes. I do appreciate the authors' point that this list is more tractable to the reader because it includes data on specific functional links rather than just a block-black statistical

posterior (though I think this argument applies primary to the underlying functional data and not to the combined scores or ranks, which are harder to interpret).

Anyway, this review is clearly not the place to draw a line in the sand on this issue. If the authors do not want to investigate the incremental value further, and the editor does not wish to push it, then I will not push any further.

Response: The reviewer makes several salient points regarding the assignment of causality to a variant and target gene identification. The aim of our approach was to make use of functional annotations to inform the identification of causal variants and their respective target genes, empowering researchers to better interpretate the results. In terms of causal variant prediction, researchers have until recently generally tended to rely solely on fine-mapping statistics to prioritise variants. In our paper we have provided functional evidence for risk variants from allele-specific transcriptional activity by performing massive parallel reporter assays (MPRA) assays, as well as ChIP-seq, ATAC-seq, and Micro-C data.

The reviewer asks why not simply go with the closest gene. While the target gene often is the closest gene, as we state in our introduction and discussion, it is not always the case. This is amply illustrated by a recent paper in Nature (doi: 10.1038/s41586-024-07501-1) in which the authors report analysis of the 21q22 risk locus for inflammatory bowel disease. As in our study, after identification of the causal variant using fine mapping and MPRA, they go on to demonstrate using Hi-C chromatin interaction data and eQTL data that neither of the closest genes (*BRWD1* or *PSMG12*) are the target gene but rather *ETS2*.

The reviewer also states why not simply use Open Target's L2G genes. However, this platform is built primarily on data from haematological tissues. Given that many predicted causal variants map to tissue specific enhancers, its applicability to contextualise variants and link these to target genes is not appropriate to derive enhancer-gene pairings for CRC risk loci (as evidenced by our demonstration of tissue specific enrichment of CRC risk loci to colonic tissue).

We do acknowledge the point that several strategies have been proposed as a model to deconvolute risk loci, but quantifying any incremental improvement is essentially intractable. The main difficulty is that there is no established definition of what a true locus-gene pair is, particularly in the non-coding genome. Even in the Open Targets Gold Standard dataset, there are only 455 curated associations. We would assert whatever the nuances of the various models, there is an increasing recognition that synthesis of multiple sources of information is the way forward - ideally, as we have sought to, by incorporating functional data from multiple modalities. We believe that this area is certainly suitable for extensive discussion in a review or editorial but feel that further discourse is outside the remit of our current paper.

#Major comment 3: Clarity of methods descriptions

The methods are improved, but I still find it very difficult to follow exactly how the variants for MPRA were picked, and I think there are still errors or confusingly written parts. I would appreciate the authors taking another pass. Specific issues to address:

- There is an inconsistency between what has been written in the point-to-point response and in the actual paper. The paper uses " $0.7 \cdot -\log_{10}(\text{P-value})$ " and the P2P uses " $P_{\text{GWAS}} < 0.7 \cdot -\log_{10}(\text{P_lead snp})$ ". Neither seems right. Are the authors trying to say " $-\log_{10}(P_{\text{GWAS}}) < 0.7 \cdot -\log_{10}(P_{\text{lead snp}})$ "?
- The use of the term "log likelihood ratio" to mean "log ratio of p-values" is confusing (as likelihood ratio has a specific meaning in hypothesis testing, which is not how the authors seem to be using it here), and should be changed for something more descriptive
- The text uses " p_{GWAS} " and " p_{variant} " - do these mean different things?
- Does " $r^2 > 0.2$ " mean " $r^2 > 0.2$ with at least one lead risk variant"?
- Does this r^2 threshold, and the " $P_{\text{GWAS}} < 10e-5$ " threshold, apply to all variants, or just the variants that meet the $0.7 \cdot \log(\text{p-value})$ criteria?

Response: We apologise for any ambiguity. To clarify matters we have revised our text accordingly. Specifically, we now state: "At each GWAS risk locus (defined by a 500kb window spanning the lead variant) we initially identified all variants with a P -value within three orders of magnitude of the lead variant P -value. As this may exclude potentially functional variants at loci where the lead variant has an especially strong association, we additionally included variants with $-\log_{10}(P_{\text{variant}}) > 0.7 * -\log_{10}(P_{\text{lead variant}})$, stipulating $r^2 > 0.2$ to the lead variant and $P_{\text{variant}} < 10^{-5}$."

In the Methods section we now state "In view of this, using data from the CRC GWAS, we selected variants for MPRA testing by first considering all variants in a 500kb window spanning each primary or conditional association (*i.e.* $\pm 250\text{kb}$ around each lead variant) that were within three orders of magnitude of the lead variant P -value. As this might not capture functional variants which remain highly significant at some loci (*i.e.* where the lead variant has an extremely strong association) we additionally considered variants having $-\log_{10}(P_{\text{variant}}) > 0.7 * -\log_{10}(P_{\text{lead variant}})$ as well as stipulating an $r^2 > 0.2$ to the lead variant and $P_{\text{variant}} < 10^{-5}$ in the GWAS."

Decision Letter, second revision:

2nd Jul 2024

Dear Dr Houlston,

Thank you for submitting your revised manuscript "Systematic prioritisation of functional variants and susceptibility genes for colorectal cancer risk loci" (NG-A63656R1). I'm delighted to inform you that we'll be happy in principle to publish it in Nature Genetics, pending minor revisions to satisfy the referees' final requests and to comply with our editorial and formatting guidelines.

Sincerely,

Safia Danovi, PhD
Senior Editor, Nature Genetics
ORCID: 0009-0007-7822-5479

Final Decision Letter:

7th Aug 2024

Dear Dr Houlston,

I am delighted to say that your manuscript "Systematic prioritisation of functional variants and effector genes for colorectal cancer risk" has been accepted for publication in an upcoming issue of Nature Genetics.

Your paper will be published online after we receive your corrections and will appear in print in the next available issue. You can find out your date of online publication by contacting the Nature Press Office (press@nature.com) after sending your e-proof corrections.

Before your paper is published online, we shall be distributing a press release to news organizations worldwide, which may very well include details of your work. We are happy for your institution or funding agency to prepare its own press release, but it must mention the embargo date and Nature

Genetics. Our Press Office may contact you closer to the time of publication, but if you or your Press Office have any enquiries in the meantime, please contact press@nature.com.

Please note that *Nature Genetics* is a Transformative Journal (TJ). Authors may publish their research with us through the traditional subscription access route or make their paper immediately open access through payment of an article-processing charge (APC). Authors will not be required to make a final decision about access to their article until it has been accepted. Find out more about Transformative Journals

Authors may need to take specific actions to achieve compliance with funder and institutional open access mandates. If your research is supported by a funder that requires immediate open access (e.g. according to Plan S principles) then you should select the gold OA route, and we will direct you to the compliant route where possible. For authors selecting the subscription publication route, the journal's standard licensing terms will need to be accepted, including [a href="https://www.nature.com/nature-portfolio/editorial-policies/self-archiving-and-license-to-publish"](https://www.nature.com/nature-portfolio/editorial-policies/self-archiving-and-license-to-publish). Those licensing terms will supersede any other terms that the author or any third party may assert apply to any version of the manuscript.

If you have not already done so, we strongly recommend that you upload the step-by-step protocols used in this manuscript to protocols.io. protocols.io is an open online resource that allows researchers to share their detailed experimental know-how. All uploaded protocols are made freely available and

are assigned DOIs for ease of citation. Protocols can be linked to any publications in which they are used and will be linked to from your article. You can also establish a dedicated workspace to collect all your lab Protocols. By uploading your Protocols to protocols.io, you are enabling researchers to more readily reproduce or adapt the methodology you use, as well as increasing the visibility of your protocols and papers. Upload your Protocols at <https://protocols.io>. Further information can be found at <https://www.protocols.io/help/publish-articles>.

Sincerely,

Safia Danovi, PhD
Senior Editor, Nature Genetics
ORCID: 0009-0007-7822-5479